# A Biomimetic Macrophage-Membrane-Fused Liposomal System Loaded with GVs-HV Recombinant Plasmid for Targeted Anti-Atherosclerosis Therapy

**DOI:** 10.3390/pharmaceutics17121618

**Published:** 2025-12-16

**Authors:** Yuelin Zhang, Wenting Gu, Kailing Yu, Qihong Chen, Hong Wang, Yinghui Wei, Hangsheng Zheng, Hongyue Zheng, Lin Liu, Fanzhu Li

**Affiliations:** 1School of Pharmaceutical Sciences, Zhejiang Chinese Medical University, Hangzhou 310053, China; 2Libraries of Zhejiang Chinese Medical University, Zhejiang Chinese Medical University, Hangzhou 310053, China; 3Department of Clinical Pharmacy, The First Affiliated Hospital of Zhejiang University, School of Medicine, Zhejiang University, Hangzhou 310003, China; 4Zhejiang Provincial Key Laboratory of Traditional Chinese Medicine for Clinical Evaluation and Translational Research, Zhejiang University, Hangzhou 310058, China; 5Key Laboratory of Neuropharmacology and Translational Medicine of Zhejiang Province, School of Pharmaceutical Sciences, Zhejiang Chinese Medical University, Hangzhou 310053, China; 6Academy of Chinese Medical Science, Zhejiang Chinese Medical University, Hangzhou 310053, China

**Keywords:** atherosclerosis, plasmid, hirudin, gas vesicle, gene delivery system

## Abstract

**Background:** Cardiovascular disease is one of the leading causes of death worldwide. The presence of atherosclerotic plaques in the arteries leads to continuous growth and obstruction of blood vessels, which ultimately leads to acute myocardial infarction and sudden cardiac death. Ultrasound-triggered GVs cavitation has great potential in plaque treatment due to its noninvasive nature and safety. **Methods**: In this work, we constructed a Hirudin–Gas Vesicle Recombinant Plasmid to achieve gene delivery using macrophage membrane/lipid membrane fusion bio-vesicles. **Results**: The bio-fusion vesicles retained the macrophage membrane protein integrin α4β1 to combine with vascular adhesion molecules highly expressed by inflammatory cells to achieve delivery; the Hirudin–Gas Vesicle Recombinant Plasmid could escape lysosomes and enter the nucleus to achieve highly efficient transfection; Hirudin and Gas Vesicles are exocytosed through cleavage peptide and exocytosis peptide, respectively; their pharmacological effects are linked and complementary. Gas vesicles can break up lesion plates with the assistance of in vitro ultrasound, and Hirudin achieves fragment ablation and anti-inflammatory and lipid regulation. **Conclusions**: GVs-HV@MM-Lipo exerts potent anti-atherosclerotic and anti-inflammatory effects with favorable safety. GVs-HV@Lipo reduces mice aortic arch plaque area by 17%, while GVs-HV@MM-Lipo+US achieves further plaque regression and improved hemodynamics. Our work opens up a new paradigm in the treatment of atherosclerosis with Chinese medicine.

## 1. Introduction

Atherosclerosis is the most common cardiovascular disease, and its main clinical manifestations are lipid accumulation and inflammation of the large arteries, which may lead to myocardial infarction and stroke [1,2,3,4]. Globally, about 18 million people die of cardiovascular diseases, mainly coronary artery and atherosclerosis, each year, accounting for more than 30% of all deaths [5]. Atherosclerosis is largely caused by elevated levels of LDL. LDL penetrates the inner walls of blood vessels to form localized accumulations, which in turn further form oxidized low-density lipoproteins (ox-LDL) [6,7]. Macrophages subsequently uptake large amounts of ox-LDL in order to remove lipids from the interior of the vessel, forming lipid droplets in their cytoplasm and turning into “foam cells” [8]. On the other hand, both damaged vascular endothelial cells and foam cells over-secrete inflammatory factors, further stimulating immune responses in the plaque microenvironment [9]. Therefore, plaque removal and localized anti-inflammation in the damaged vessel lining is the preferred treatment option for atherosclerosis.

Traditional Chinese medicine has unique advantages in regulating the inflammatory microenvironment and dissolving plaque [10,11]. Hirudin, the active ingredient in the Chinese animal medicine leech, is a highly efficient and specific thrombin inhibitor [12]. Hirudin is widely used in the treatment of cardiovascular diseases because of its potent anticoagulant and thrombolytic properties [13,14]. Studies have shown that hirudin has a protective effect on the vascular endothelium, reversing endothelial dysfunction and delaying the progression of atherosclerosis [15]. Hirudin can also inhibit the expression of inflammatory factors through the PARs-p38-NF-κB pathway and can inhibit the expression of the chemotactic inflammatory factor monocyte chemotactic protein (MCP-1), which in turn inhibits the inflammatory response and exerts an anti-atherosclerotic effect [16]. In addition, the enzymatic digest of leech inhibited foam cell formation [13]. These studies have shown that hirudin has a promising role in the prevention and treatment of atherosclerosis. However, hirudin still suffers from a short in vivo half-life (≈2 h in humans) and poor stability, requiring frequent high-dose administration to maintain efficacy—this not only reduces patient compliance but also increases the risk of systemic bleeding due to non-targeted distribution [17]. Moreover, free hirudin cannot actively accumulate at atherosclerotic lesions, leading to low local drug concentrations and limited ability to modify plaque progression. These unmet needs highlight the value of developing a targeted, sustained delivery system—such as plasmid-based expression—to optimize hirudin’s therapeutic potential while addressing its half-life and targeting deficiencies.

Achieving plaque ablation or volume reduction in a noninvasive manner promises a breakthrough in clinical care. Noninvasive ultrasound-triggered GVs cavitation has great potential based on the mechanical and cavitation effects of ultrasound [18]. Experiments have shown [19] that focused ultrasound can selectively destroy tissues lacking the normal collagen and elastic fiber skeleton, such as thrombi and atherosclerotic plaques while avoiding damage to vascular. Gas Vesicles (GVs) are expressed primarily in aquatic photosynthetic organisms and are structured as amphiphilic cylindrical protein shells that allow for the free exchange of gases and the exclusion of liquid water, resulting in a thermodynamically stable air vesicle structure [20]. Since the interior of GVs is filled with gas, they scatter acoustic waves, generate ultrasonic contrast, and exhibit high sensitivity and safety—attributed to their natural protein structure that ensures biodegradability and low immunogenicity. Moreover, engineered multigene clusters encoding GVs can be heterologously expressed in bacterial and mammalian cells to enable gene expression [21], laying a foundation for integrating GVs with gene delivery systems to realize ultrasound-triggered, targeted plaque intervention.

Plasmid-based gene delivery—ideal for combining with GVs due to its low immunogenicity, high safety, and flexible editing [22,23]—can express target proteins at lesions and enable fusion expression of different sequences (e.g., GVs and therapeutic factors) to retain original properties while achieving synergistic effects. In the field of gene delivery, liposomes are the preferred carriers, yet they lack abundant surface receptors compared to natural cells, resulting in weak targeting and limited accumulation at atherosclerotic plaques [24,25]. Macrophages, as key immune cells, are highly relevant to atherosclerosis: most accumulate in plaque necrotic cores, influence disease progression, and serve as potential therapeutic targets [26,27,28]. Thus, we designed a novel strategy for atherosclerosis modulation: preparing macrophage membrane-liposome fusion vesicles (GVs-HV@MM-Lipo) loaded with GVs-HV recombinant plasmid (Figure 1). The fusion system uses macrophage membrane integrin α4β1 to actively target inflamed endothelium through α4β1/VCAM-1 binding [29,30], while the GVs-HV plasmid enables secretion of GVs for ultrasound-triggered plaque physical fragmentation and hirudin for anticoagulation, thrombolysis, and inflammation modulation. This integrated platform achieves targeted plaque clearance and local inflammation regulation, opening a new administration mode for animal-derived drugs and providing new ideas for TCM-based atherosclerosis therapy.

## 2. Materials and Methods

### 2.1. GVs-HV Recombinant Plasmid Construction and Verification

The gas vesicle–Hirudin recombinant plasmid was constructed by Genechem (Shanghai, China). In a typical experiment, the recombinant genes for gas vesicles (GVs), hirudin (HV), and luciferase were ligated to the cloning vector GV126 with a signal peptide in the presence of restriction endonuclease. The genes with signal peptides were ligated to the cloning vector GV126 by restriction endonuclease sites in the presence of ligase and treated with alkaline phosphatase to prevent self-intercrossing of the genes. and treated with alkaline phosphatase to prevent self-ligation of cloning vector GV126. The ligation product was transformed into *E. coli* by the heat shock method. Recombinant plasmids were extracted using a plasmid extraction kit. The recombinant plasmid DNA was sequenced using an automated sequencer. The recombinant plasmid was identified by 1% agarose gel electrophoresis and the corresponding restriction endonuclease. The recombinant plasmid was identified with the appropriate restriction endonuclease. The recombinant gene was subcloned into the expression vector GV126 to construct the eukaryotic secretory expression vector of GVs-HV recombinant plasmid. The GVs-HV recombinant plasmid was used as a control and the corresponding primers were used for plasmid PCR identification.

HUVEC cells were seeded in 6-well plates at a cell density of 20 × 104 cells per well and incubated overnight at 37 °C in a 5% carbon dioxide incubator to make the cells adhere. The lipo2000/GVs-HV complex (3:1, *w*:*w*) was prepared, diluted with serum-free DMEM (Gibco, CA, USA) in proportion, added to the plate, and placed in the incubator for further incubation. After 6 h, the culture medium was washed away, the fresh medium containing 10% serum was replaced after PBS (Gibco) washing, and the culture was continued for 20 h. The cell supernatant was collected, and the ultrasound and non-ultrasound control groups were set up. The two samples were placed onto a copper grid, dyed with 2% ammonium molybdate, air-dried at room temperature, and then observed using a transmission electron microscope (TEM) at an acceleration voltage of 200 kV.

### 2.2. Macrophage Membrane Isolation

A hypotonic lysis method was chosen to isolate the Raw 264.7 membrane [31]. Briefly, cells were collected through centrifugation. Then washed with PBS and resuspended with 4 °C precooled 10 mM Tris-HCl, and placed on ice for 1 h followed by squeezing for about 45 min on ice with a cell homogenizer. The ratio of 10 mM Tris-HCl (Shanghai Macklin Biochemical Co., Ltd., Shanghai, China) to glucose was 3:1 by adding 1 M glucose solution. After that differential centrifugation was used to separate the membrane: first, 2000× *g*, 4 °C, centrifuging for 10 min, the supernatant was collected to centrifuge with 3000× *g*, 4 °C for 30 min. The precipitate was resuspended with 10 mM Tris-HCl to glucose (3:1), followed by centrifuging with 3000× *g*, 4 °C for 30 min again. The precipitate was dispersed with HEPES at pH 7.3–7.4 and stored at −80 °C for subsequent experiments.

### 2.3. Verification of Lipid Membrane and Macrophage Membrane Fusion

To visualize membrane fusion, a Förster resonance energy transfer (FRET)-based dye pair (DiO and DiI; Beyotime, Shanghai, China) was used to label the lipid membrane and macrophage membrane, respectively. Laser Scanning Confocal Microscopy (CLSM, Zeiss LSM880, Oberkochen, Germen) was employed for sample imaging to further confirm the fusion process.

### 2.4. Preparation of GVs-HV@Lipo

DOTAP (10%; Avanti, VA, USA), PC S100 (Avanti, VA, USA), DSPE-PEG2000 (Avanti, VA, USA), and cholesterol (Avanti, VA, USA) were dissolved in chloroform (Sinopharm Chemical Reagent Co., Ltd., Beijing, China) at a mass ratio of 4:1:1. The organic solvent (chloroform) was evaporated via thin-film evaporation to form a thin lipid film, which was then vacuum-dried overnight to ensure complete removal of residual chloroform. The GVs-HV plasmid was diluted in pH 7.4 PBS and added to the hydrated liposomes at 60 °C for 1 h, with a liposome-to-plasmid mass ratio of 1:40.

### 2.5. Preparation and Characterization of GVs-HV@MM-Lipo

After the preparation of GVs-HV@Lipo, the lipid solution was mixed with macrophage solution and extruded through a membrane filter (pore size: 200 nm). The hydrodynamic diameter and zeta potential of GVs-HV@MM-Lipo were evaluated using Dynamic Light Scattering (DLS, Zetasizer Nano S90, Malvern Panalytical Ltd., Malvern, UK). Blank liposomes, GVs-HV@Lipo and GVs-HV@MM-Lipo were placed on a copper grid, stained with 2% ammonium molybdate, air-dried, and observed. GVs-HV@lipo and GVs-HV@MM-Lipo were placed in 10% serum. Particle size and potential were measured after 0, 0.5, 1, 3, 6, 12, 24, 36, 48 and 72 h and serum stability was observed.

### 2.6. Cellular Uptake of GVs-HV@Lipo and GVs-HV@MM-Lipo

Cell models for in vitro validation were created to mimic the internal environment of atherosclerotic inflammation. HUVECs (BNCC378266, BNCC^®^, Henan, China) were cultured overnight with LPS (Beyoyime, Shanghai, China) to induce inflammatory endothelium. In addition, ox-LDL (20 μg/mL) (Beijing Solarbio Science & Technology Co., Ltd., Beijing, China) and LPS (1 μg/mL) were added to the medium and incubated with endothelial cells for 24 h to induce lipid-overloaded endothelium. the PBS group was only replaced with fresh medium without any stimulating factors. Similarly, mouse macrophage Raw 264.7 cells (BNCC354753, BNCC^®^, Henan, China) were incubated with LPS-containing medium for 24 h to induce lipid-overloaded macrophages, i.e., foam cells. The PBS group was only replaced with fresh medium. Dio-labeled GVs-HV@Lipo and GVs-HV@MM-Lipo were added to the medium to the different cell groups mentioned above and cultured in an incubator, and then the unabsorbed fluorochromes were washed with PBS. After washing the fluorophores, cells were digested with trypsin to obtain cell suspension. After centrifugation, the medium was replaced with PBS and the fluorescence intensity was detected and quantified by flow cytometry.

DiO, a dye that emits a bright green fluorescence when bound to phospholipid bilayers, was applied to GVs-HV@Lipo and GVs-HV@MM-Lipo, respectively, through a hydration process to observe the uptake of the cells over time. The uptake of HUVECs and Raw264.7 cells was observed by CLSM at four time points: 1, 3, 6, and 9 h in different groups.

### 2.7. In Vitro Lysosomal Escape Ability of GVs-HV@MM-Lipo

The efficacy of nucleic acid drugs is predicated on the ability to escape the lysosome, so Lysosome Tracker Red (Beyoyime, Shanghai, China) and Cy5-DNA (GENEWIZ, Suzhou, China) were chosen to verify that a macrophage-membrane-fused drug-delivery system could safely deliver nucleic acids to the vicinity of the nucleus. Fluorescence intensity was detected and quantified by CLSM.

### 2.8. Transfection Efficiency of GVs-HV@MM-Lipo

HUVECs and inflammatory endothelial cell models were constructed as described previously. Plasmid transfection efficiency was tested using the Firefly luciferase reporter gene assay kit (Beyoyime, Shanghai, China). Using a 96-well plate suitable for chemiluminescence detection, an appropriate amount of effector cells was inoculated in each well. Cells were treated with GVs-HV@MM-Lipo and incubated at different times. In addition, cells were treated with different masses of GVs-HV@MM-Lipo, and wells without cells were used as negative controls. After incubation, an appropriate amount of firefly luciferase reporter gene cell lysate was added to the cell culture medium. After complete lysis, centrifugation was performed and the supernatant was taken for assay.

The process followed these steps: Take the sample, add an appropriate amount of Firefly luciferase assay reagent that has been equilibrated to room temperature, and mix well. Incubate at room temperature (25 °C) for 5 min and perform chemiluminescence detection using a multifunctional microplate reader with chemiluminescence detection.

### 2.9. Expression of Gas Vesicles In Vitro

HUVECs and foam cells were co-cultured in Transwell plates to mimic the inflammatory environment of atherosclerosis. Briefly, endothelial cells were cultured in the upper chamber, foam cells were cultured in the lower chamber, and GVs-HV@MM-Lipo was cultured in the upper chamber for 6 h. The dissolved oxygen content in the cell supernatant before and after ultrasound was detected by a dissolved oxygen detector. Meanwhile, the PBS-treated cells were set as the control group.

In addition, scanning electron microscopy (SEM, SU8010, Hitachi, Japan) was performed to observe the effects of GVs-HV@MM-Lipo and GVs-HV@MM-Lipo+US on the morphology of foam cells. After transfection of GVs-HV recombinant plasmid into HUVEC cells and secretion, the intracellular structure of foam cells before and after sonication was observed by transmission electron microscopy (TEM).

### 2.10. In Vitro Lipid-Lowering and Anti-Inflammatory Ability of GVs-HV@MM-Lipo

Addition of ox-LDL and LPS to the medium and incubation with Raw 264.7 cells for 24 h induced lipid-overloaded macrophages, i.e., foam cells, as described previously. Oil red O and hematoxylin staining (Beijing Solarbio Science & Technology Co., Ltd., Beijing, China) were used to identify lipid droplets in foam cells.

Transwell plates were used to co-culture HUVECs and foam cells to mimic the inflammatory environment of atherosclerosis, as previously described. Cell supernatants from the upper and lower lumen were collected separately for the detection of cellular inflammatory factors and chemokines, including TNF-α, IL-1β, IL-10, MCP-1, IFN-γ, iNOS, VEGF-A, and α-SMA (NanJing JianCheng Bioengineering Institute, Nanjing, China).

### 2.11. In Vitro Cytotoxicity Assay of GVs-HV@MM-Lipo by MTT Assay

HUVECs and Raw 264.7 cells were seeded in 96-well plates at a density of 6 × 103 per well and cultured overnight for attachment. A higher concentration of GVs-HV@MM-Lipo (0.1, 0.2, 0.4, 0.8, 1.6, 3.2, 6.4, 12.8 μg, quantified with GVs-HV recombinant plasmid) was added to the medium and incubated for another 6 h. Medium containing MTT (10%) was added to each Petri dish and incubated for 4 h after the end of incubation. Dimethyl sulfoxide was used instead of the above medium and the optical density (OD) value at 490 nm was recorded on a microplate reader. The relative cell viability was calculated by the following formula:(1)$$Relative\;cell\;viability(\%)=\frac{{OD}_{sample}-{OD}_{blank}}{{OD}_{control}-{OD}_{blank}}\times 100\%$$

### 2.12. Ethics Statement

Six-week-old male ApoE^−/−^ mice were purchased from SLAC Laboratory Animal Co. Ltd. (Shanghai, China). This study obtained ethical approval from the Institutional Review Board. All experiments were performed under experimental protocols approved by The Tab of Animal Experimental Ethical Inspection of the First Affiliated Hospital, Zhejiang University School of Medicine (SCXK2019-0002). The environmental conditions in the mouse facility were 12 h light/dark cycle, the temperature was around 22 ± 2 °C, relative humidity range of 50 ± 10%, and all mice were free access to food and water.

### 2.13. GVs-HV@MM-Lipo Relieves Plaque Symptoms In Vivo

The six-week-old male ApoE^−/−^ mice were randomly divided into 6 groups, with 9 mice in each group. According to different treatment regimens, tail vein injections were administered once every 2 days for a total of 30 days. The treatment regimens were as follows: the blank control group received normal saline injections, the positive drug control group received monotherapy lovastatin injections (0.3125 mg/kg), the hirudin drug group received recombinant hirudin solution injections (10 mg/kg), the GVs-HV@Lipo group received GVs-HV@Lipo solution injections (with a GVs-HV recombinant fusion gene mass of 10 μg), the GVs-HV@MM-Lipo group received GVs-HV@MM-Lipo solution injections (with a GVs-HV recombinant fusion gene mass of 10 μg), and the GVs-HV@MM-Lipo+US group received GVs-HV@MM-Lipo solution injections combined with external ultrasound (1 MHz, 150 W, 3 sec on, 2 sec off, 20% amplitude, for 1 min). Injections were administered with the assistance of a visual tail vein injection fixator, and the body weight of the mice was recorded after each administration.

One month after treatment, all mice were executed. The aortic arches of the atherosclerotic mice were isolated and removed. After removing the surrounding fatty tissue, it was placed in a fixative for more than 24 h, then the tissue was removed from the fixative and rinsed twice with PBS. The longitudinal portion of the vessel was carefully cut along the vessel wall with dissecting scissors. The dissected vessels were immersed in oil red staining solution for 60 min at 37 °C and then removed. The fatty plaques in the lumen were differentiated into orange or bright red with 75% ethanol. After rinsing twice with distilled water, photographs were taken and quantified.

### 2.14. In Vivo Anti-Atherosclerosis and Anti-Inflammatory Study of GVs-HV@MM-Lipo

The Vevo Small Animal Ultrasound Cardiac Imager is used to provide integrated diagnostic and therapeutic ultrasound support. Mice were selected for isoflurane during imaging and anesthesia was maintained at 1–2%. They were first anesthetized in an induction box and then placed supine on a physiological information monitoring platform. Hair near the heart site was removed with depilatory cream and then washed with water before image acquisition. On the one hand, the aortic arch section was an adjusted right parasternal section, and the MS400 probe incision was directed toward the animal’s lower jaw; the probe was slightly rotated clockwise to optimize the image to observe the peak aortic arch and aortic blood flow. On the other hand, placing the MS400 probe arm against the right side of the animal and tilting the object bed slightly to the left brought the right ventricle into the imaging field of view, allowing observation of blood flow, ejection fraction, and ventricular wall thickness.

Aortic arches and roots of mice subjected to different treatments were taken and frozen sections were stained with oil red O for examination of aortic plaques. Paraffin sections of aortic arch were used for H&E staining and examined immunohistochemically with CD68 and α-SMA antibodies. In addition, cellular inflammatory factors and chemokines, including α-SMA, VEGF-A, iNOS, TNF-α, IL-1α, MCP-1, IFN-γ, IL-10, and IL-1β, were detected.

Besides, anti-inflammatory capacity of GVs-HV@MM-Lipo was tested by measuring cholesterol (CHO) and low-density lipoprotein cholesterol (LDL-C).

### 2.15. In Vivo Biosafety Profile of GVs-HV@MM-Lipo

After one month of treatment, all ApoE^−/−^ mice were put to death. Heart, liver, spleen, lungs, and kidneys were collected for HE staining. Blood and serum were collected for pathological and biochemical studies. Routine blood counts, especially white blood cells, platelets, and red blood cells, were also used as auxiliary indices.

### 2.16. Statistical Analysis

All data shown are mean ± s.d. One-way analysis of variance (ANOVA) followed by the Bonferroni post-test was used for data with 3 groups. Statistical graphs are plotted by GraphPad Prism 10 (Version 10.4.0, MA, USA). Detailed analysis methodology and *p* values can be found in the figure legends. *p* values smaller than 0.05 were considered statistically significant.

## 3. Results

### 3.1. Construction and Verification of GVs-HV Recombinant Plasmid

First, we designed and constructed the GVs-HV recombinant plasmid (Figure 2A). The entire recombinant plasmid contained two major protein sequences, Gas Vesicle, and Hirudin, and also incorporated the luciferase gene for in vivo visualization. We inserted Igk exocytosis signal peptides at the front of the hirudin and gas vesicle sequences, respectively, to enable the plasmid to be distributed for expression and secreted out of the cell. Meanwhile, the isolated expression of hirudin and gas vesicles was achieved by the cut peptide T2A on the sequence.

The GVs-HV recombinant plasmid is 7707 bp in length and contains a 1257 bp insertion target fragment. The recombinant plasmid was enzymatically identified with MluI and HindIII, and the electrophoresis results were as expected (Appendix A), indicating successful digestion. The product was amplified by real-time PCR, and the size of the product was 1257 bp. It was identified by 1% agarose gel electrophoresis, and the result was as expected (Appendix A). The positive clones were singled out for bacterial PCR identification, and the PCR product of the positive transformants was 472 bp in size, as expected (Figure 2B). The identified positive transformants were sequenced and the sequencing results were compared with the target gene sequences. The results showed that the sequenced sequences were identical to the target sequences. The above results indicated that the GVs-HV recombinant plasmid designed by us was successfully constructed.

In order to verify whether the GVs-HV recombinant plasmid was correctly expressed and its functionality, we collected the supernatant of transfected cells and observed it under the stimulation of sonication. Compared with the unsonicated group, the supernatant of the sonicated group contained obvious bubble-like structures (Figure 2C), which was attributed to the exocytosis of the bubble-containing fusion proteins into the cell supernatant, cavitation under sonication activation, and collapse of the cell secretion, resulting in bubble-like structures. In addition to verifying GVs expression via TEM-observed bubble-like structures, hirudin expression was indirectly confirmed by its pharmacological effects and the plasmid’s design—hirudin is co-expressed with GVs via T2A cleavage and secreted via IgK signal peptide, with successful plasmid transfection (Figure 3M) ensuring its expression alongside functional GVs.

### 3.2. Construction and Characterization of GVs-HV@MM-Lipo

Next, we prepared GVs-HV@Lipo by thin-film hydration and coextruded it with macrophage membranes obtained by differential centrifugation, resulting in GVs-HV@MM-Lipo (Figure 2D). The screening of the proportions of different components in the prescription is shown in Appendix A. The optimal fusion ratio of the two membranes was screened according to the effect of fluorescence resonance energy transfer on the subsequent liposome preparation. The results showed that the emission peaks of the two fluorescent dyes appeared simultaneously when the liposomes (DiO) were fused with macrophage membranes (DiI) at different ratios, indicating that the proximity between the two membrane materials inserted with different dyes would produce different degrees of FRET effects (Figure 2E). The fluorescence emission peak of hybrid vesicles near 550 nm increased with the increase in macrophage membrane content. When the ratio of liposome to macrophage membrane was 2:1, the fluorescence intensity of DiO was the lowest and that of DiI was the highest, indicating that the FRET effect was the strongest at this time and the fusion effect was the best.

Figure 2F shows the particle size and surface potential of MM, free lipid, GVs-HV@lipo, and GVs-HV@MM-Lipo. It can be seen that the particle size of GVs-HV@Lipo did not change much compared to free lipids. While the particle size of GVs-HV@MM-Lipo was about 250 nm. It was shown that MM was more electronegative while free lipid was positively charged and GVs-HV@Lipo was electroneutral, which might be due to the insertion of phospholipid bilayer by part of the plasmid. The potential of GVs-HV@MM-Lipo was about −10 mV, which was intermediate between MM vesicles and free liposomes, and also indicated that the membrane fusion was successful. TEM showed that free liposomes had a pearlescent rounded structure, with an average diameter of less than 200 nm (Figure 2(Gc)), whereas both GVs-HV@Lipo (Figure 2(Gb)) and GVs-HV@MM-Lipo (Figure 2(Ga)) had an irregular sphere with a fingerprinted structure. The above can indicate that the preparation of GVs-HV@MM-Lipo was successful (Figure 2(Ga)). We further investigated the stability of GVS-HV@MM-Lipo. We chose 10% serum PBS (PH = 7.4) to simulate the in vivo environment. The changes in particle size and potential of GVs-HV@Lipo and GVs-HV@MM-Lipo were measured under this condition as shown in Appendix A. After 72 h, the particle sizes of the two internal vesicles did not change much, and both of them exhibited good stability.

### 3.3. Uptake and Intracellular Transport of GVs-HV@MM-Lipo

Liposomes are highly susceptible to phagocytosis by the mononuclear phagocyte system after entering the bloodstream, leading to their rapid clearance [32,33]. We observed the uptake efficiency of GVs-HV@MM-Lipo by endothelial HUVECs and Raw 264.7, as well as the uptake trend under inflammatory conditions. Firstly, we found by flow cytometry that macrophage-membrane-fused liposomes were more readily phagocytosed by endothelial cells under inflammatory conditions (Figure 3A,B). By fluorescence semi-quantitative analysis, the uptake of GVs-HV@MM-Lipo by endothelial cells was approximately twice that of macrophages (Figure 3C,D). This aspect may be due to the natural inflammatory targeting mechanism of MM, i.e., its membrane integrin α4β1 specifically recognizes VCAM-1 on the surface of inflamed endothelial cells and is responsible for the active targeting of GVs-HV@MM-Lipo. By CLSM, we observed that the green fluorescence signal of GVs-HV@Lipo and GVs-HV@MM-Lipo gradually increased between 1–6 h and gradually declined at 9 h during co-incubation with HUVECs. It was characterized by significant time-dependent absorption (Appendix A). And the signal was brighter in the GVs-HV@MM-Lipo group. This indicated that in HUVECs, GVs-HV@MM-Lipo fused with the cell membrane had better access to the cell interior (Figure 3E,F). In contrast, the uptake of GVs-HV@Lipo or GVS-HV@MM-Lipo by macrophages was much less efficient than that by endothelial cells (Figure 3G,H and Appendix A). We speculate that this may be due to the predominance of HUVECs due to their membrane fusion mechanism. For macrophages, phagocytosis predominates, leading to a further reduction in uptake efficiency.

Thereafter we analyzed by lysosomal staining. As can be seen in Figure 3I, DiO-stained GVs-HV@MM-Lipo hardly overlapped with lysosomes of endothelial cells. The results in Figure 3J showed that DiO-stained GVs-HV@MM-Lipo partially overlapped with lysosomes of macrophages. Co-localization analysis using Image J revealed that the lysosomes of GVs-HV@MM-Lipo co-localized further away from endothelial cells compared to macrophages, suggesting that it is more likely that GVs-HV@MM-Lipo enters into HUVEC cells through membrane fusion, or that GVs-HV@MM-Lipo is better able to escape from the capture of endothelial cell lysosomes [34]. Our observation that DNA in bio-vesicles was more distributed in the nucleus compared to free DNA and free liposomes also suggests that macrophage membranes fused with lipid membranes to deliver recombinant plasmids may have a higher expression efficiency (Figure 3K). Therefore, we examined the transfection efficiency of GVs-HV@MM-Lipo by the expression of plasmid fusion luciferase (Figure 3L). Appendix A shows the screening of the appropriate number of transfected plasmids in different cell groups. The amount of plasmid in each cell was controlled at 0.16 μg, and the fluorescein intensity was observed by different transfection times. As shown in Figure 3M, the fluorescence intensity was stronger after transfection of HUVEC with GVs-HV@MM-Lipo. Compared with Raw 264.7, the RLU intensity was about twice as strong in the HUVECs group.

### 3.4. GVs-HV@MM-Lipo Mediated Linkage of Hirudin and Gas Vesicles

In the early stages of atherosclerosis, endothelial cell injury is a common phenomenon, followed by an increase in endothelial permeability, and small amounts of lipids can invade. As the disease progresses, large numbers of monocytes and smooth muscle cells proliferate, and macrophages phagocytose lipids to form foam-like cells. We used Transwell plates to simulate the lesion microenvironment (Figure 4A)—specifically, GVs-HV-transfected HUVECs were seeded in the upper chamber and ox-LDL-induced foam cells in the lower chamber—to test the effect of GVs-HV@MM-Lipo on foam cells. In this system, GVs and hirudin secreted by the transfected HUVECs in the upper chamber can diffuse to the lower chamber and act on foam cells, which is reflected by subsequent Oil Red O staining results (detailed in the following analysis of Figure 4A) indicating inhibited lipid accumulation in foam cells. Firstly, we collected cell supernatants after GVs-HV@MM-Lipo treatment and then applied with ultrasound, and it could be found that the dissolved oxygen content of the group given GVs-HV@MM-Lipo combined with ultrasound was 15 mg/L, which was about 1.5 times that of the group without ultrasound, while the PBS group presented relatively less dissolved oxygen (Figure 4B). This was due to the exocytosis of gas vesicles into the supernatant; these vesicles were triggered by sonication to produce a cavitation effect, thereby generating oxygen [35,36].

SEM showed that transfected HUVEC exocytosed gas vesicle–Hirudin fusion proteins and affected the foam cells (Figure 4C, upper). Foam cells in the untreated group had regular spherical surfaces, and sonication alone did not affect cell morphology, whereas foam cells treated with GVs-HV@MM-Lipo+US showed a concave structure, suggesting that the inertial cavitation of gas vesicles can produce a strong mechanical effect, making the cell structure flat [37]. Further, as observed by TEM (Figure 4C, below), the untreated foam cells had a complete cellular structure, there was obvious white lipid droplets distributed in the cytoplasm. After ultrasonication, the foam cell structure was basically unchanged. In contrast, after giving GVs-HV@MM-Lipo+US treatment, the cell structure became flattened, the edge was not clear, and there was a clear vacuole-like structure in the cell and a tendency of autophagy. This was due to the cavitation effect, which made the cell membrane more permeable to the drug by physically damaging the cell structure. Meanwhile, hirudin promoted the autophagy process, inhibited inflammation and oxidative stress, and suppressed the activation of NLRP3 inflammatory vesicles and PINK-1/Parkin pathway [38].

In addition, we examined the behavior of GVs-HV@MM-Lipo in reducing intracellular lipid droplets in foam cells (Figure 4D). The red color was essentially absent in PBS-treated macrophages; whereas large red areas were visible in macrophages from the ox-LDL-only group, suggesting that macrophages are effective in ox-LDL uptake. After GVs-HV@MM-Lipo administration, the red-stained areas were significantly reduced (Figure 4E); TEM/SEM observations of lipid droplet depletion in this context are supported by prior studies, which demonstrated that effective anti-atherosclerotic interventions (e.g., targeted lipid regulators) reduce foam cell lipid content, as characterized by decreased lipid droplet size and number under electron microscopy [39]. This reduction may be attributed to hirudin regulating intracellular lipid deposition by modulating foam cell lipid metabolism, thereby improving the pathological microenvironment of atherosclerosis and inhibiting disease progression.

On the other hand, proinflammatory cytokines or chemokines play an important role in the atherosclerotic process, and we investigated the levels of cytokines induced or not by ox-LDL in HUVECs (Figure 4G and Appendix A) and Raw 264.7 (Figure 4F and Appendix A), respectively. Representative proinflammatory cytokines (TNF-α and IL-1β) and chemokines (MCP-1) within the HUVEC and Raw 264.7 groups were significantly elevated in the ox-LDL-induced group, whereas they showed a steady decrease in the GVs-HV@MM-Lipo and ultrasound treated groups. This may be a result of MM conferring GVs-HV@MM-Lipo with the ability to neutralize inflammatory cytokines2 and synergize with hirudin to exert anti-inflammatory and lipid-modulating effects. Appropriate activation of the IFN-γ signaling pathway promotes macrophage activation, while iNOS, a hallmark of classically activated macrophages [40,41], may be the result of proinflammatory cytokine stimulation. In HUVECs (Figure 4G), both oxLDL-induced and GVs-HV@MM-Lipo-treated groups showed an increasing trend in IFN-γ, with the oxLDL group having a more significant upward trend; in Raw 264.7 foam cells (Figure 4F), no such increasing trend in IFN-γ was observed in the GVs-HV@MM-Lipo-treated group. α-smooth muscle actin (α-SMA) and vascular endothelial growth factor-A (VEGF-A) are markers of the vascular endothelium [42], which are associated with the proliferation of vascular endothelial-producing cells, and are indicators capable of maintaining lesion stability. Compared with the trend of increasing levels seen in the ox-LDL-induced group, the GVs-HV@MM-Lipo-treated group showed the same stable state as normal cells, suggesting that atherosclerosis may be stabilized and prevented from further deterioration after GVs-HV@MM-Lipo treatment. In addition to this, we also examined the cytotoxicity of GVs-HV@MM-Lipo towards HUVECs and Raw 264.7. We found no significant effect of the formulations on the viability of either cell (Appendix A). Hemolysis experiments similarly demonstrated the safety of the GVs-HV@MM-Lipo (Appendix A).

### 3.5. GVs-HV@MM-Lipo Relieves Plaque Symptoms and Exhibits Anti-Inflammatory Effects

To verify the ability of GVs-HV@MM-Lipo to attenuate atherosclerotic symptoms in mice, we established an atherosclerosis model with male ApoE−/− mice on a high-fat diet for 10 weeks. Depending on the group, tail vein injections were administered every 2 days for 30 consecutive days of treatment (Figure 5A). At the end of treatment, we isolated and stripped the aortic arches of the mice and stained them for oil red O (Figure 5B). The quantitative oil red O plot (Figure 5C) showed that the red area of each treatment group was smaller than that of the saline group, with the hirudin group and the GVs-HV@MM-Lipo+US group showing better results, and the red area of the aortic arch was about 50% of that of the saline group. Figure 5D shows the comparison of aortic arch blood flow before and after treatment. It can be found that GVs-HV@MM-Lipo+US has a better therapeutic effect, with an increase in aortic arch blood flow, indicating a gradual decrease in the plaque area in the vessel. This suggests that gas vesicles can have a cavitation effect on plaques. During the treatment period, the left ventricular septal thickness, the left intraventricular diameter, and the left ventricular posterior wall thickness were all reduced more significantly. The relief of left ventricular hypertrophy reduced the pressure on the heart and reduced the compression on the coronary arteries, allowing the ventricles to fill normally (Figure 5E,G).

This result was also supported by the Vevo Small Animal Ultrasound Cardiac Imager, which showed changes in aortic arch flow (Figure 5H and Appendix A) and LV wall thickness (Figure 5I and Appendix A) before and after treatment. Specifically, Figure 5H shows representative echocardiographic images of aortic arch blood flow in mice before and after treatment—compared with pre-treatment, the GVs-HV@MM-Lipo+US group exhibits clearer, more uniform blood flow signals and increased peak flow velocity. These changes indicate reduced aortic arch plaque obstruction (consistent with Oil Red O staining results in Figure 5B,C) and improved vascular patency, which not only corroborates the increased blood flow observed in Figure 5D but also directly reflects the formulation’s ability to restore normal aortic hemodynamic function, further validating its anti-atherosclerotic efficacy.

We then performed HE staining and oil red O staining of the aorta, and the red spots showed the location of the plaques, as shown in Figure 6A. By examining the intimal surface lesions of the aorta, we found that the saline-treated group had the largest plaque area, accounting for almost 29% of the entire aortic tissue area (Figure 6B). The plaque area was reduced by approximately 17% in the GVs-HV@Lipo treatment group, which is attributed to the sustained anti-inflammatory and lipid-lowering effects of hirudin expressed by the GVs-HV plasmid. In addition, the treatment effect of GVs-HV@MM-Lipo+US was slightly better than that of GVs-HV@MM-Lipo. We also compared the positive drugs lovastatin and hirudin simultaneously and found that the area of plaque and necrotic cores was significantly reduced in the GVs-HV@MM-Lipo+US group, which provided better relief of symptoms of intravascular plaque occlusion.

Monocytes are recruited to the lesion site early in the plaque formation process [43,44]. Animal models have shown that Ly6CHI proinflammatory monocytes rapidly enter plaques and induce proinflammatory cytokine-mediated endothelial cell activation. Meanwhile, macrophages differentiated from monocytes were actively involved in cholesterol accumulation and promoted plaque formation by maintaining a proinflammatory microenvironment. This suggests that macrophages and monocytes play important roles in atherosclerosis [45]. Effective control of them can prevent atherosclerosis. We found that the GVs-HV@MM-Lipo+US group significantly reduced the number of macrophages and monocytes in aortic arch plaques (Figure 6A,C), which suggests that GVs-HV@MM-Lipo can moderately slow down the process of atherosclerosis.

During atherosclerosis, smooth muscle cells proliferate abnormally and secrete collagen. Increased collagen content contributes to plaque stabilization and leads to vessel narrowing [46]. Analysis of anti-α-SMA staining (Figure 6A,D) showed that the number of positive cells was relatively high in the saline group, accounting for approximately 6% of the lumen area. Lovastatin reduced the number of positive cells on the plate by 4%. The positive cell content in the GVs-HV@MM-Lipo group, including the ultrasound-assisted group, was approximately 3%. This suggests that in our therapeutic strategy, due to the anti-inflammatory and lipid-droplet-modulating pharmacological effects of hirudin (positivity rate of 3% in the hirudin group), there was a reduction in local inflammation within the vasculature and a reduction in further stimulation of smooth muscle cells by the inflammatory milieu.

As far as proinflammatory factors are concerned, TNF-α plays a crucial role in disrupting the macrovascular and microvascular circulations. The expression of TNF-α increases the production of ROS, leading to endothelial dysfunction in many pathophysiological states [47]. Relevant animal experiments have shown that atherosclerosis-susceptible mice lacking IL-1β have reduced atherosclerotic plaque load, whereas mice exposed to excess IL-1β have increased plaque load. IL-6 improves risk prediction of atherosclerosis and inflammation in patients with cardiovascular risk. IL-1α is the supreme activator of inflammatory responses, responding to external disturbances, and initiating and maintaining inflammatory processes. The above proinflammatory factors showed a decreasing trend during treatment in all groups, with greater anti-inflammatory capacity in the GVs-HV@MM-Lipo+US group (Figure 6E and Appendix A). In terms of chemokine (MCP-1), the GVs-HV@MM-Lipo+US group showed a significant decreasing trend compared to the positive control group, which may be due to the fact that GVs-HV@MM-Lipo inherited the specific TNFR-2 receptor on MM, which isolates inflammation [31]. Many other cytokines associated with atherosclerosis showed similar results to the in vitro experiments. IL-10, an anti-atherosclerotic cytokine, was slightly elevated in the preparation group, which may be related to the recruitment of more leukocytes to counteract the inflammatory environment. Indicators related to neovascularisation and intravascular pressure, such as iNOS, VEGF, and α-SMA, were correspondingly lower, which may be attributed to the reduction in intravascular plaque area during treatment, resulting in less vascular compression, which in turn reduced the expression of these factors (Figure 6F and Appendix A).

In addition, the body weight of mice in all groups was relatively stable during the treatment period (Appendix A). We performed safety tests on the vital organs of the mice in each group (Appendix A), which also showed that GVs-HV@MM-Lipo had a favorable biosafety profile. In serum biochemical tests, lipid indices such as cholesterol (CHO, Appendix A) and low-density lipoprotein (LDL-C, Appendix A) showed a decreasing trend after treatment. The results of routine blood tests showed that the white blood cell counts of mice in the GVs-HV@MM-Lipo group showed a slow decreasing trend and gradually converged to the normal indices, whereas the platelet and erythrocyte indices showed no significant differences among the groups (Appendix A).

Notably, the GVs-HV@MM-Lipo+US group showed a trend of slightly reduced plaque area compared to the GVs-HV@MM-Lipo group (Figure 6B), though the difference was not statistically significant. This trend may reflect ultrasound-triggered cavitation enhancing local hirudin bioavailability, which is consistent with the in vitro dissolved oxygen data (Figure 4B). The lack of significant in vivo differences highlights the need for parameter optimization to amplify GVs’ mechanical effects while ensuring safety.

## 4. Discussion

In this study, a composite delivery platform based on GVs-HV@MM-Lipo was proposed, which combined the anticoagulant and anti-inflammatory effect of hirudin with the acoustic response characteristics of gas vesicle protein, and achieved significant therapeutic effect in the ApoE^−/−^ mouse atherosclerosis model. The experimental results showed that GVs-HV@Lipo reduced the plaque area of the aortic arch by about 17%, while the GVs-HV@MM-Lipo+US group further improved the hemodynamic parameters. In the quantitative analysis of oil red O, the red staining area of the aortic arch in the hirudin and GVs-HV@MM-Lipo+US groups was about half of that in the control group. Meanwhile, the serum inflammatory factors TNF-α, IL-1β and MCP-1 were significantly decreased, the vascular remodeling-related index α-SMA was decreased, and the anti-inflammatory factor IL-10 was slightly increased. This evidence, together, indicates that the platform can achieve plaque volume reduction, inflammatory microenvironment improvement and blood flow function recovery in vivo, and the weight, organ pathology and hematological indexes suggest good safety.

A critical distinction between the GVs-HV@MM-Lipo platform and free hirudin lies in addressing the latter’s inherent limitations while enabling synergistic therapeutic effects—all supported by our existing data. First, in terms of pharmacokinetic optimization: Free hirudin’s short half-life requires frequent administration, but our platform uses a recombinant plasmid to drive sustained hirudin expression and secretion at the lesion site (mediated by the IgK signal peptide, Figure 2A; nuclear localization of DNA, Figure 3K). This design ensures continuous local availability of hirudin, avoiding the ‘peak-trough’ concentration fluctuations of free hirudin and reducing systemic exposure risks. Second, in terms of targeting efficiency: Free hirudin distributes systemically, but GVs-HV@MM-Lipo achieves active targeting to inflamed endothelium via α4β1/VCAM-1 binding—flow cytometry shows that its uptake by inflammatory HUVECs is 2-fold higher than that of non-targeted liposomes (Figure 3A–C), and confocal imaging confirms enhanced intracellular accumulation (Figure 3E,F). This targeted delivery directly increases hirudin’s local concentration at plaques, as reflected by more significant reduction in foam cell lipid droplets (Figure 4D,E) compared to free hirudin. Third, in terms of therapeutic breadth: Free hirudin acts primarily via anticoagulation and anti-inflammation, but our platform integrates three functions: macrophage membrane-mediated immune regulation (downregulates MCP-1, Figure 6E), ultrasound-triggered GVs cavitation (enhances local bioavailability, Figure 4B), and sustained hirudin-mediated lipid modulation. Together, these functions result in better plaque stabilization (reduced α-SMA+ cells, Figure 6D) and hemodynamic recovery (improved aortic arch blood flow, Figure 5D) than free hirudin—confirming the platform’s unique value beyond simple drug delivery.

Hirudin has been shown to have anticoagulant and anti-inflammatory effects in previous studies, such as reducing lipid droplet accumulation in foam cells, inhibiting the release of inflammatory factors and thrombosis [48,49]. The results of this study further showed that the hirudin plasmid delivery was more effective in reducing macrophage infiltration in the aortic arch than free hirudin, and the secretion efficiency and safety were optimized by inserting the secretion signal peptide and the exogenous sequence, thereby enhancing the anti-plaque efficacy in vivo. Gas vesicles are a class of genetically encoded hollow protein nanostructures with unique acoustic properties, which can be used as ultrasound contrast agents and can also promote the release and uptake of genes and drugs thru cavitation effects [50,51,52,53,54]. In this study, GVs and hirudin sequences were independently expressed thru the T2A cleavage peptide and secreted by the IgK signal peptide. We speculate that GVs may enhance the efflux and local availability of hirudin under the action of ultrasound, resulting in sound-triggered synergistic drug efficacy. This hypothesis still needs to be further established by quantifying the hirudin secreted before and after ultrasound, analyzing the integrity and cavitation threshold of GVs, and verifying the autophagy and inflammatory pathway.

The synergistic linkage between GVs cavitation and hirudin pharmacology constitutes the core innovative mechanism of our platform. Our data provide clear evidence for the formation of GVs and their ultrasound-triggered cavitation activity. TEM imaging of cell supernatants confirmed the presence of hollow, cylindrical nanostructures characteristic of GVs post-sonication (Figure 2C). Functionally, this cavitation event was quantified by a significant increase in dissolved oxygen levels (Figure 4B), a hallmark of inertial cavitation. This cavitation effect enhances therapeutic efficacy through two interconnected mechanisms. Firstly, it exerts a direct mechanical force on atherosclerotic plaques and foam cells. The SEM and TEM images of foam cells treated with GVs-HV@MM-Lipo+US clearly show structural deformation, membrane disruption, and vacuolation (Figure 4C), indicative of physical damage induced by cavitation-generated shockwaves and microjets. This mechanical action directly contributes to plaque fragmentation and volume reduction, as corroborated by the reduced plaque area and improved aortic arch blood flow in vivo (Figure 5B–D,H). Secondly, and perhaps more critically, the cavitation-induced membrane disruption serves as a powerful physical sensitization strategy. By increasing cell membrane permeability, it significantly enhances the penetration and bioavailability of the co-expressed and secreted hirudin, thereby potently amplifying its anti-inflammatory and lipid-lowering pharmacological effects (Figure 4D–G). Thus, the GVs and hirudin are not merely co-expressed; their effects are functionally linked, with cavitation priming the pathological site for enhanced pharmacotherapy.

Rationale for ultrasound trigger and GVs stability in vivo: Our strategy is specifically designed to overcome the potential instability of conventional exogenous contrast agents during systemic circulation. Unlike pre-formed microbubbles, our platform enables the localized and sustained expression of GVs directly within the target endothelial cells at the atherosclerotic site. This “situ synthesis and secretion” approach ensures a fresh supply of GVs precisely where the intervention is needed, mitigating concerns about their degradation during long blood circulation. The subsequent application of external ultrasound provides a noninvasive, spatiotemporally controlled switch to activate these pre-delivered GVs on demand. This allows us to maximize the therapeutic effect at the optimal time while minimizing off-target effects. The observed trends of superior hemodynamic improvement and inflammatory reduction in the GVs-HV@MM-Lipo+US group (Figure 5D,H and Figure 6E,F), albeit without a statistically significant difference in final plaque area compared to the non-ultrasound group (Figure 6B), validate the biological relevance of this approach. The modest additional benefit from ultrasound in this specific model may be attributed to two factors: (i) The ultrasound parameters were deliberately chosen with a primary emphasis on biosafety, potentially limiting the full mechanical cavitation effect on deeper aortic plaques. (ii) The potent and sustained anti-inflammatory and lipid-regulating effects of locally expressed hirudin (evident in the GVs-HV@MM-Lipo group) may have partially masked the incremental contribution of the physical cavitation effect. Future studies will focus on optimizing the ultrasound parameters (intensity, duration, pulse sequences) within a defined safety window to fully harness the synergistic potential of this mechano-pharmacological therapy.

Notably, the membrane vesicle modification of GVs-HV@MM-Lipo provided advantages in targeting and immunomodulation. The experiments showed that the downregulation of MCP-1 and the reduction in α-SMA may be related to the α4β1/VCAM-1 binding mediated by the macrophage membrane and the TNFR-2-related inflammatory isolation effect. In the future, α4β1/VCAM-1 blocking and TNFR-2 neutralization experiments can be designed to clarify the contribution of each factor to the overall efficacy.

Compared with the existing therapeutic strategies for atherosclerosis, the advantage of this study lies in the realization of a triple intervention mechanism: ① macrophage membrane modification brings active targeting (via α4β1/VCAM-1 binding) and immune microenvironment regulation; ② ultrasound-triggered GVs exert a cavitation effect that enhances local bioavailability of secreted hirudin (supported by in vitro dissolved oxygen increase, Figure 4B), which is a safe and noninvasive way to promote drug efficacy; ③ hirudin confers multiple pharmacological effects of anticoagulation, anti-inflammation, and lipid-lowering (Figure 4E and Figure 6B). This ‘targeted delivery-physical enhancement-pharmacological therapy’ ternary coupling provides a new comprehensive solution for plaque reduction and stabilization.

At the level of clinical transformation, it is still necessary to pay attention to the optimization of ultrasound parameters and cavitation safety window to avoid the risk of plaque rupture; evaluate the safety of hirudin-related coagulation; monitor the immunogenicity and antibody production of GVs; and conduct long-term follow-up and large animal studies to verify the long-term efficacy and clinical feasibility. If these problems can be solved, the GVs-HV@MM-Lipo platform is expected to become a new strategy for precise treatment of atherosclerosis and image-treatment integration.

## 5. Conclusions

In summary, we successfully constructed a novel Gas Vesicle–Hirudin recombinant plasmid and designed a macrophage membrane/lipid membrane fusion gene delivery system for atherosclerotic plaque removal and inflammation regulation. GVs-HV recombinant plasmid gene entered the inflammatory endothelial cells through the macrophage membrane, escaped the lysosome into the nucleus, and successfully expressed and secreted gas vesicles and hirudin, realizing the linkage of the two pharmacological effects. The gas vesicles are activated by ultrasound in vitro, exerting a cavitation effect and physically fragmenting plaques, while hirudin exerts anticoagulant thrombolytic and lipid-droplet-modulating effects. The two collaborate to further down-regulate the levels of inflammatory factors to achieve plaque removal and inflammation modulation. In future studies, we may further compare the therapeutic effects of separate plasmids of hirudin and gas vesicles and conduct studies on the mechanism of hirudin in the treatment of atherosclerosis. This work is the first time that gas vesicles have been used in gene therapy, which lays the foundation for the subsequent integration of gas vesicles in diagnosis and treatment and provides new ideas and inspiration for the use of traditional Chinese medicine in the treatment of atherosclerosis.

## Figures and Tables

**Figure 1 pharmaceutics-17-01618-f001:**
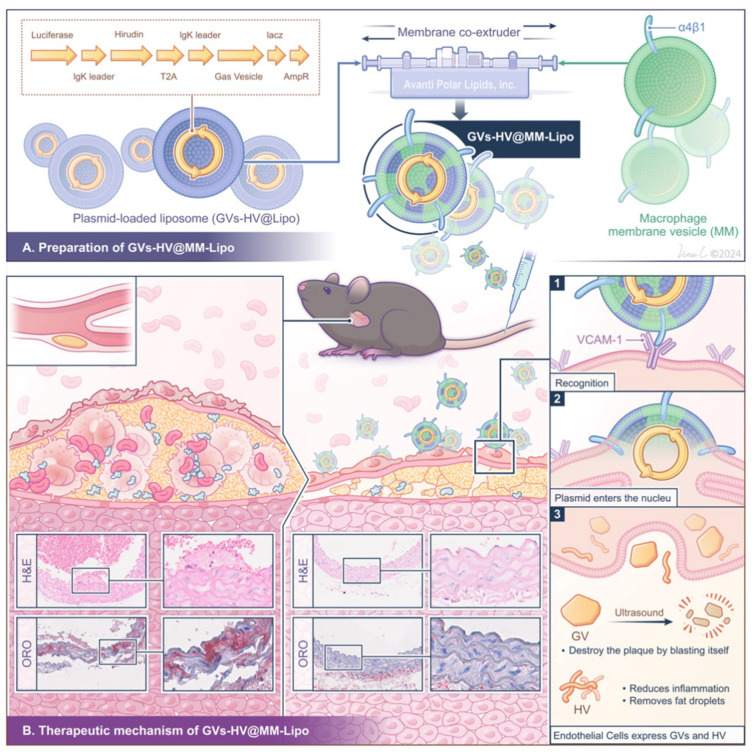
A schematic representation of the preparation and the anti-atherosclerosis mechanism of GVs-HV@MM-Lipo in a mouse model.

**Figure 2 pharmaceutics-17-01618-f002:**
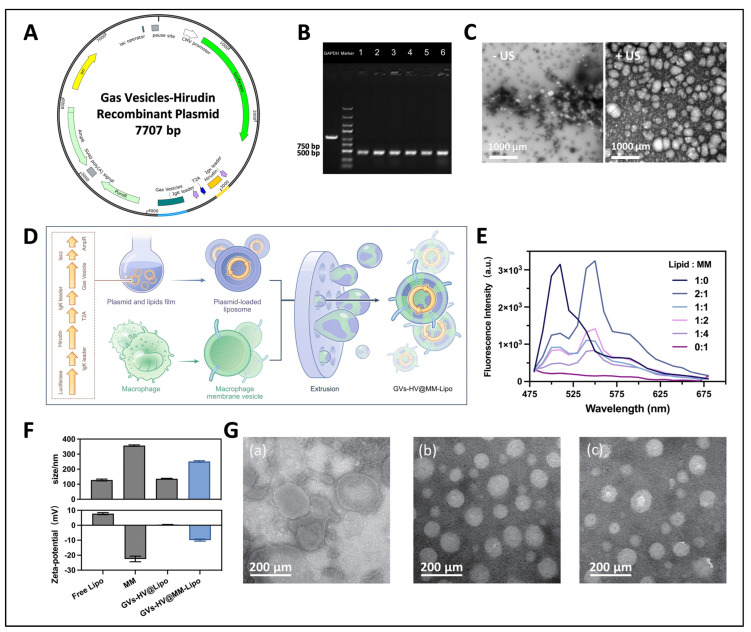
Construction and characterization of GVs-HV@MM-Lipo. (**A**) Schematic diagram of GVs-HV recombinant plasmid. (**B**) Results of PCR identification of GVs-HV recombinant plasmid fragments. The first list is positive control (GAPDH), the second list is Marker, and numbers 1–6 represent transformants. (**C**) TEM images of cell supernatants from before and after sonication. (**D**) Schematic diagram of GVs-HV@MM-Lipo preparation. (**E**) FRET effect to examine the optimal ratio of lipid membrane fusion with macrophage membrane. (**F**) Size and zeta potential distribution of Free lipo, MM, GVs-HV@MM-Lipo, GVs-HV@Lipo (mean ± s.d., *n* = 3). (**G**) Representative TEM images of GVs-HV@MM-Lipo (**a**), GVs-HV@Lipo (**b**), and Free Lipo (**c**).

**Figure 3 pharmaceutics-17-01618-f003:**
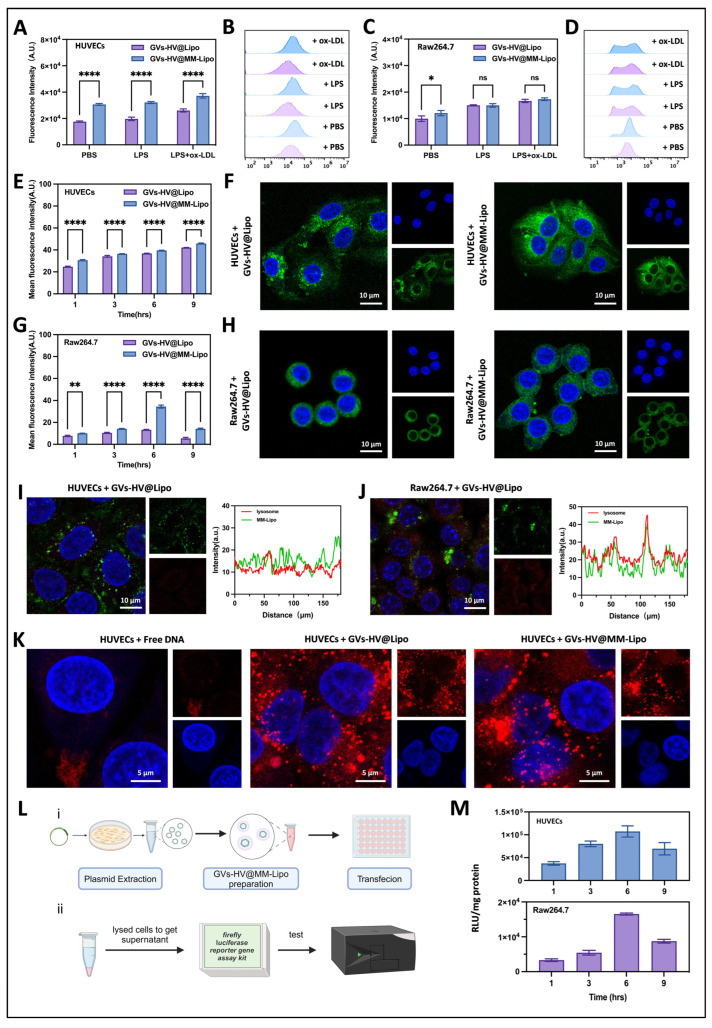
In vitro cellular uptake, lysosomal escape ability and transfection efficiency evaluation of GVs-HV@MM-Lipo. (**A**–**D**) Flow cytometry analysis of HUVECs (**A**,**B**) and Raw 264.7 (**C**,**D**) for uptake of GVs-HV@MM-Lipo or GVs-HV@Lipo under different conditions. (**E**–**H**) Semi-quantitative transient uptake fluorescence of HUVECs (**E**) and Raw 264.7 (**G**) on GVs-HV@MM-Lipo or GVs-HV@Lipo (mean ± s.d., *n* = 3), and their confocal representative graphs after 6 h of co-incubation (F for HUVECs, H for Raw 264.7). (**I**,**J**) CLSM characterization of GVs-HV@MM-Lipo co-localization with lysosomes and its fluorescence co-localization analysis in HUVECs and Raw 264.7 cells. The co-localizations of whole image of I&J were performed with Image J. (**K**) CLSM characterizes the ability of different carrier-loaded sample DNA to localize to the nucleus. (**L**) Schematic diagram of plasmid transfection (**i**) and luciferase assay (**ii**). (**M**) Fluorescence semi-quantification of plasmid transfection efficiency over time (RLU/mg protein) (*n* = 6). All data are expressed as mean ± s.d. Statistical analysis was performed by one-way ANOVA. ns means not significant, * *p* ≤ 0.05, ** *p* ≤ 0.01, and **** *p* ≤ 0.0001.

**Figure 4 pharmaceutics-17-01618-f004:**
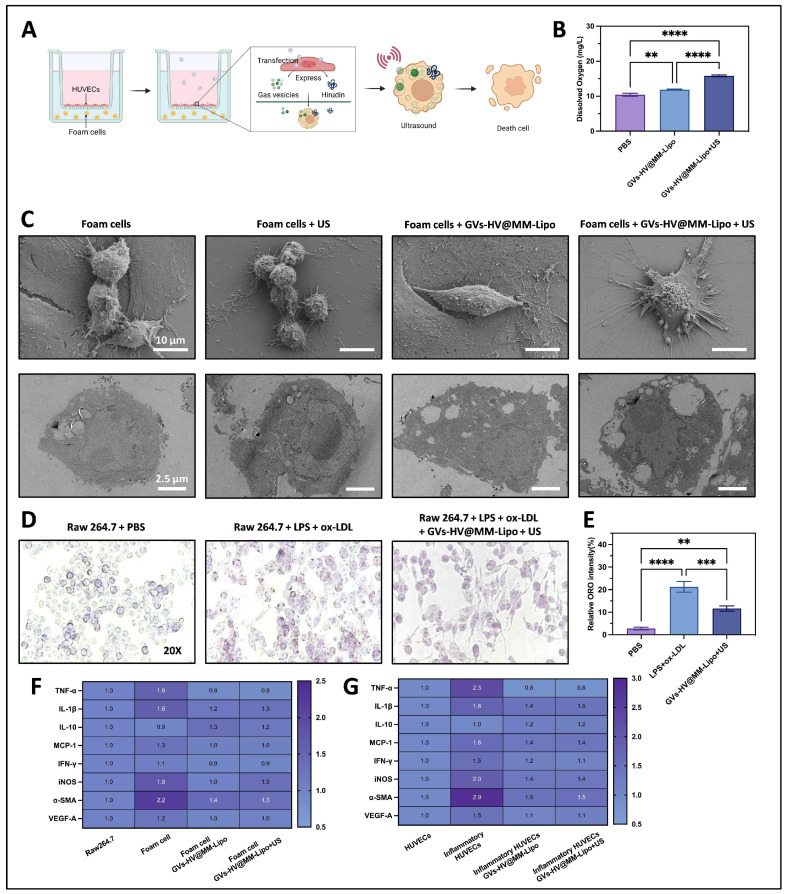
In vitro evaluation of anti-inflammatory and lipid-lowering capacity of GVs-HV@MM-Lipo. (**A**) Schematic diagram of the experimental procedure. Transwell plates were utilized to culture HUVECs in the upper chamber and foam cells in the lower chamber to establish an in vitro atherosclerotic inflammatory environment. (**B**) Comparison of extracellular dissolved oxygen after treatment in different groups (*n* = 3). (**C**) Morphological and intracellular changes in foam cells observed by SEM and TEM, respectively. (**D**,**E**) Representative microscopic images and semi-quantifications of oil red O-stained cells. (**F**,**G**) Cytokine levels in cell supernatants from different groups of treated cells (*n* = 3). All data are presented as mean ± s.d. Statistical analysis was conducted using one-way ANOVA. ** *p* ≤ 0.01, *** *p* ≤ 0.001, and **** *p* ≤ 0.0001.

**Figure 5 pharmaceutics-17-01618-f005:**
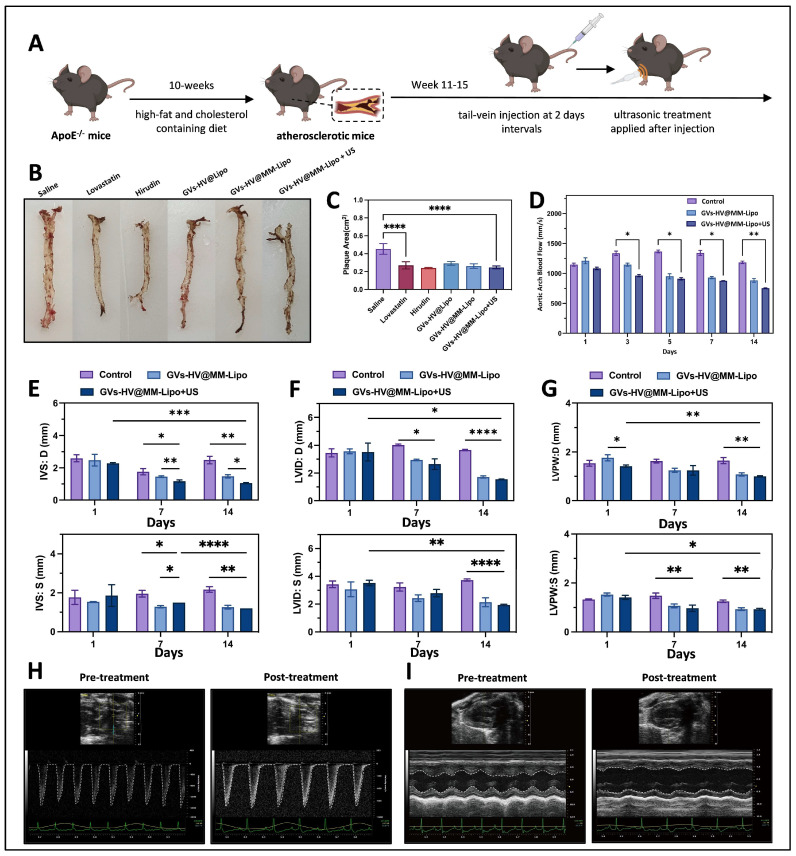
In vivo evaluation of anti-atherosclerotic capacity of GVs-HV@MM-Lipo. (**A**) Schematic diagram of the establishment of atherosclerotic mice and the treatment cycle. (**B**) Results of oil red O staining of aorta macrosomes from atherosclerotic mice. (**C**) Quantitative analysis of aortic tissue lesion area (*n* = 5). (**D**) Evaluation of aortic arch blood flow before and after treatment. (**E**–**G**) Evaluation of diastolic left ventricular septal thickness, systolic left ventricular septal thickness (**E**), left ventricular end-diastolic internal diameter, left ventricular end-systolic internal diameter (**F**), left ventricular end-diastolic posterior wall thickness, and left ventricular end-systolic posterior wall thickness (**G**) before and after treatment, where diastole is D and systole is S. (**H**) Echocardiographic characterization of aortic arch blood flow before and after treatment. (**I**) Echocardiographic characterization of left ventricular wall thickness before and after treatment. All data are presented as mean ± s.d. Statistical analysis was conducted using one-way ANOVA. * *p* ≤ 0.05, ** *p* ≤ 0.01, *** *p* ≤ 0.001, and **** *p* ≤ 0.0001.

**Figure 6 pharmaceutics-17-01618-f006:**
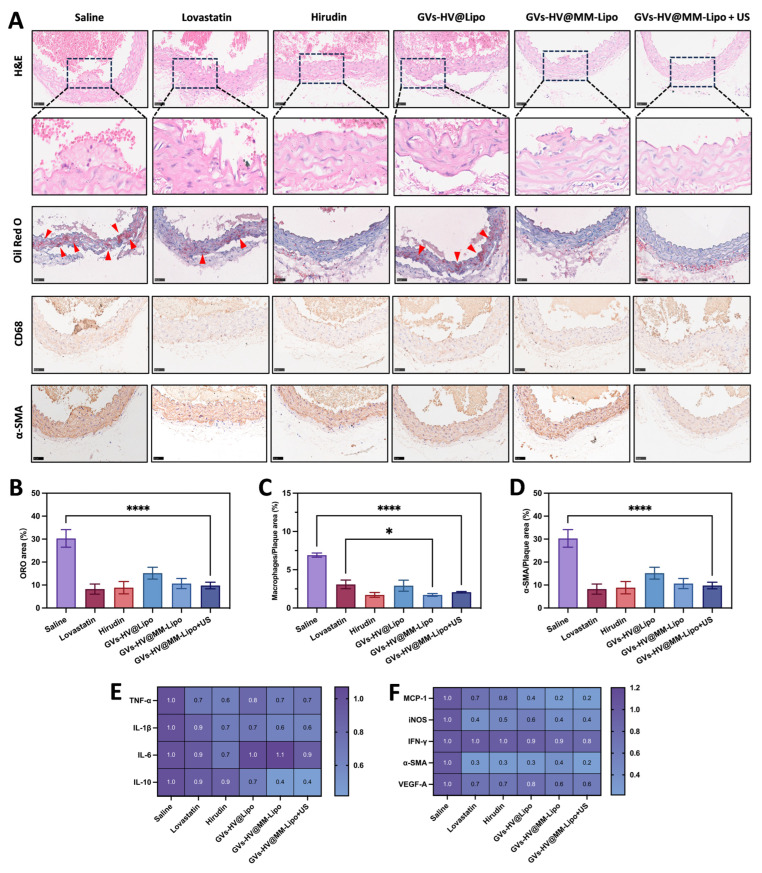
Evaluation of plaque relief and anti-inflammatory capacity of GVs-HV@MM-Lipo. (**A**–**D**) Representative photographs and quantitative analysis of H&E, Oil Red O, CD68, and α-SMA-stained aortic tissues. Scale Bar = 50 μm. (**E**,**F**) The levels of proinflammatory and chemotactic cytokines in serum from different groups of treated groups. All data are presented as mean ± s.d. Statistical analysis was conducted using one-way ANOVA. * *p* ≤ 0.05, and **** *p* ≤ 0.0001.

## Data Availability

The raw data supporting the conclusions of this article will be made available by the authors on request.

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
