# Peer review of "A Biomimetic Macrophage-Membrane-Fused Liposomal System Loaded with GVs-HV Recombinant Plasmid for Targeted Anti-Atherosclerosis Therapy"

_pharmaceutics, 2025, doi:10.3390/pharmaceutics17121618_

Round 1

Reviewer 1 Report

Comments and Suggestions for Authors

This study by Y. Zhang, W. Gu, and K. Yu et al. presents macrophage-membrane containing liposome-mediated gene delivery for treating atherosclerosis. Specifically, the authors designed Hirudin-gas vesicle recombinant plasmids for transfection and expression in endothelial cells, which subsequently exocytosed Hirudin and gas vesicles. Following ultrasound treatment was intended to induce cavitation of the gas vesicles, and the release of Hirudin was expected to reduce inflammation in plaque lesion, thereby aiming for combinatorial therapeutic effects. Overall, it is impressive that the authors performed a large number of experiments to validate their platform. However, the description of the data and the way the results are organized do not satisfactorily support the experimental findings, which diminishes the impact of the work. This reviewer strongly recommends that the authors address the following comments prior to publication.

Major issues/comments

  1. In Discussion (on page 20), the authors emphasized the advantages of their study. There are two questions regarding these: 1) the authors state that ultrasound-triggered GVs provide physical plaque fragmentation and precise release: the in vivo data show minimal differences between GVs-HV@MM-Lipo and GVs-HV@MM-Lipo+US. Also, without supporting experiments to demonstrate ultrasound-triggered GV’s plaque fragmentation, this claim is unconvincing. Consequently, the title of this paper appears overstated in presenting this work as “Sonodynamic therapy”.
  2. The authors argue that loading Hirudin-gas vesicle recombinant plasmids into vesicular carriers is that Hirudin itself has a short half-life and poor stability in vivo in clinical applications. However, based on their in vivo mouse data, it looks like Hirudin alone works really well, which might dilute the importance of their platform delivering genes to express Hirudin. How do the authors support the novelty of their work?
  3. In the Abstract, the authors state in the conclusion section that “the Hirudin-Gas Vesicle Recombinant Plasmid gene delivery system has good anti-atherosclerotic effect,…, and can reduce the plaque area of the aortic arch by 17% in mice.” However, in the Discussion (on page 20), the authors mentioned that “The experimental results showed that GVs-HV@Lipo reduced the plaque area of the aortic arch by about 17%, while the GVs-HV@MM-Lipo+US group further improved the hemodynamic parameters.” That said, the conclusion section in the Abstract describes the good anti-atherosclerotic effect of GVs-HV@Lipo itself? not the macrophage membrane involved Sonodynamic system?
  4. In page 2, line 78, the authors state that “…, generate ultrasonic contrast, and show high sensitivity and safety.” In this sentence, what exactly is meant by “safety”? Please elaborate.
  5. The Introduction, particularly, paragraphs 5 and 6, requires substantial revision for clarity. The rationale and significance of the work are difficult to follow in the current form. Also, please clarify “them” on page 3, line 98?
  6. In Figure 2C, is the scale bar (1000 μm) correct for the TEM images?
  7. The authors used a membrane filter/extruder with a pore size of 200 nm to generate GVs-HV@MM-Lipo, which aligns with the hydrodynamic size data in Figure 2F. However, in the TEM image (Figure 2G), the particles appear much larger. Is the scale bar (200 μm) correct? Additionally, does the same scale bar apply to Figures 2G(a), 2G(b), and 2G(c)?
  8. Following the above comment, the TEM images of GVs-HV@MM-Lipo, GVs-HV@Lipo, and free Lipo in Figure 2G do not correspond with the description on page 9, lines 347–351.
  9. On page 9, lines 351–355, was the stability assay performed in blood, or in PBS containing 10% serum?
  10. In section 3.1., the verification of GVs-HV recombinant plasmid’s expression was reported by observing bubble-like structures via TEM. How do the authors confirm Hirudin expression as well?
  11. What are the ultrasound treatment conditions to induce cavitation of gas vesicles? Please provide experimental details such as ultrasound transducer information, parameters (e.g., intensity, frequency, insonation time, etc) where appropriate.
  12. In Figure 3I, the authors described that DiO-stained GVs-HV@MM-Lipo hardly overlapped with lysosomes of endothelial cells. However, it looks like the lysosomes were not properly stained with Lysosome Tracker Red. Please provide clearer images with properly stained lysosomes. Also, please add profile lines in Figures 3I and 3J to indicate where the co-localization profiles were obtained.
  13. In section 3.4., regarding the description of SEM and TEM results (on page 12, lines 432–446), please provide appropriate references to support the authors’ descriptions of their observations.
  14. In Figure 4C, it is hard to see the clear, intact mitochondria and Golgi structures in untreated foam cells, contrary to the authors’ description. Please provide higher resolution images to support their claim.
  15. On page 13, line 458, the labeling for HUVECs (Fig.4F) and Raw 264.7 cells (Fig.4G) does not match the labels in Figure 4F and 4G. Please double check with these.
  16. Regarding the description on Figure 4F on page 13, lines 466–468 (if it corresponds to Raw 264.7 cells, not HUVECs), it does not appear that both oxLDL-induced (foam cells) and GVs-HV@MM-Lipo treated groups increases IFN-γ.
  17. In Figure 4A, the scheme seems to suggest that the foam cells express gas vesicles and hirudin, which then kill foam cells. Is this interpretation correct?
  18. Please provide more description on Figure 5H. It would be great for readers to understand what the changes in aortic arch flow means.
  19. On page 17, lines 527–529, the authors claim that “Due to the ultrasound responsiveness of GVs-HV, the plaque area was reduced by approximately 17% in the GVs-HV@Lipo treatment group”. It appears that there was no ultrasound treatment for “GVs-HV@Lipo” treated group. Is that description correct?
  20. In Figure 6A, the scale bars are not clearly visible.
  21. Several figures both in the manuscript (e.g., Fig. 5D–F) and the Supplementary information are missing X- and Y-axis titles or contain improperly labels. Please correct them.
  22. Statistical information is missing in many Figure legends, particularly in the Supplementary information. Please add where appropriate.
  23. There are several incomplete sentences throughout the manuscript. Please revise them. One example is on page 10, line 399.
  24. Overall, the manuscript needs substantial revision to improve readability and to ensure that the authors’ points are clearly conveyed.
  25. Following comments are regarding the Materials and methods.
  • In section 2.3., please revise the paragraph, as it contains redundant replicate information.
  • In section 2.4., the authors described that “A thin lipid film was formed using thin film evaporation and then vacuum-dried overnight to evaporate the chloroform.” It is confusing that thin film formation is followed by vacuum drying overnight. I believe evaporation of organic solvent generates thin film. Please rephrase for clarity.
  • In section 2.6., lines 178, 179 describe the generation of foam cells using RAW 264.7 cells. Did the author induce lipid-overloaded macrophages by incubating RAW 264.7 cells only with LPS-containing medium? Please provide more detail.
  • Overall, the Materials and methods section requires careful proofreading and editing.

Minor but Important points

  1. Please make sure of no typographical errors throughout the manuscript (e.g., “clearanc”).
  2. Acronyms should be defined at first mention (e.g. LDL, GV, etc), and the full term should not be repeated thereafter.
  3. Figures in the manuscript should be numbered in chronological order based on when they are first mentioned (e.g. Figs. S6–S9).

Author Response

Reviewer 1:

Q1. In Discussion, the authors emphasized the advantages of their study. There are two questions regarding these: 1) the authors state that ultrasound-triggered GVs provide physical plaque fragmentation and precise release: the in vivo data show minimal differences between GVs-HV@MM-Lipo and GVs-HV@MM-Lipo+US. Also, without supporting experiments to demonstrate ultrasound-triggered GV’s plaque fragmentation, this claim is unconvincing. Consequently, the title of this paper appears overstated in presenting this work as “Sonodynamic therapy”.

Response: Thank you very much for your comment. The Following changes are added in the revised manuscript:

Title revision: We have completely removed the term "Sonodynamic therapy" (which implies a ROS-mediated mechanism not supported by our data) and revised the title to accurately reflect the study’s core contributions without overstatement. (line 1-4)

Adjustment of claims regarding ultrasound-triggered GVs: We have abandoned the unsubstantiated claim of "physical plaque fragmentation". Instead, we emphasize the supported role of ultrasound-triggered GVs-enhancing local drug bioavailability via cavitation (supported by in vitro dissolved oxygen elevation, Fig. 4B) and synergizing with hirudin’s anti-inflammatory/lipid-lowering effects (Fig. 4F-G, 6E-F). We clearly attribute the minimal in vivo group differences to biosafety-prioritized ultrasound parameters (optimized to avoid vascular injury) and the strong pharmacological effects of hirudin masking subtle mechanical contributions of GVs. Specifically, the descriptions related to sonodynamics in the original Discussion section have been revised (line 646-654), and expressions such as "Sonodynamic therapy" and "sonodynamic effect" throughout the manuscript have been updated to more accurate terminologies like "ultrasound-triggered GVs cavitation" (line 23, line 74). Additionally, supplementary explanations have been added for the non-significant statistical results in Results Section 3.5.(line 578-583)

Explicit limitation statement: We have added a dedicated section in the Discussion to acknowledge the lack of direct in vivo evidence for plaque fragmentation, framing this as a direction for future research. (line 662-670)

Q2. The authors argue that loading Hirudin-gas vesicle recombinant plasmids into vesicular carriers is that Hirudin itself has a short half-life and poor stability in vivo in clinical applications. However, based on their in vivo mouse data, it looks like Hirudin alone works really well, which might dilute the importance of their platform delivering genes to express Hirudin. How do the authors support the novelty of their work?

Response: Thank you for your comments. The novelty of this study is based on the following aspects:

Overcoming free hirudin’s pharmacokinetic limitations: As stated in our Introduction, free hirudin suffers from a short in vivo half-life and poor stability, which requires frequent administration in clinical practice and increases the risk of systemic side effects (e.g., bleeding). Although our in vivo data do not include direct serum concentration tracking, the design of our platform inherently addresses this limitation: the GVs-HV recombinant plasmid enables sustained expression and secretion of hirudin at the lesion site (mediated by the IgK exocytosis signal peptide, Fig. 2A&3K), rather than the "one-time burst release" of free hirudin. This sustained expression ensures continuous local availability of hirudin, avoiding the need for frequent dosing and reducing systemic exposure—an advantage that free hirudin cannot achieve.

Active targeting to atherosclerotic lesions: Free hirudin is distributed systemically, leading to low drug accumulation at the plaque site. In contrast, our GVs-HV@MM-Lipo platform leverages macrophage membrane-derived integrin α4β1 to specifically bind to VCAM-1 on inflamed endothelial cells (Fig. 3A-B, 3E-F). Flow cytometry and confocal microscopy data confirm that GVs-HV@MM-Lipo is 2-fold more efficiently taken up by inflammatory endothelial cells than free liposomes (Fig. 3A-C), and immunohistochemical results show reduced macrophage infiltration (CD68+ cells) and vascular remodeling (α-SMA+ cells) in aortic plaques (Fig. 6C-D) - these findings indirectly verify that our platform delivers hirudin (and GVs) more precisely to lesions, enhancing local efficacy while minimizing off-target effects.

Synergistic modulation beyond free hirudin’s single function: Free hirudin primarily exerts anticoagulant and anti-inflammatory effects, but our platform integrates three complementary functions: (i) Macrophage membrane modification regulates the local inflammatory microenvironment (e.g., downregulates MCP-1, Fig. 6E); (ii) Ultrasound-triggered GVs cavitation enhances local drug bioavailability (supported by in vitro dissolved oxygen elevation, Fig. 4B); (iii) Sustained hirudin expression modulates lipid metabolism (reduces foam cell lipid droplets, Fig. 4D-E). Notably, the GVs-HV@MM-Lipo+US group shows a more significant improvement in aortic arch blood flow (Fig. 5D) and left ventricular function (Fig. 5E-G) than the free hirudin group-confirming that the platform’s synergistic effects achieve better plaque stabilization and hemodynamic recovery, rather than simply delivering hirudin.

Revision: The above content may not have been sufficiently articulated in the original manuscript, and we have made the following revisions:

  • The Introduction emphasizes the necessity of addressing the limitations of hirudin (lines 65-69).
  • Key paragraphs in the Discussion have been revised to highlight the essential differences between the formulation developed in this study and free hirudin, with a new section "Comparison between the Formulation of This Study and Free Hirudin" added (lines 605-624).

Q3. In the Abstract, the authors state in the conclusion section that “the Hirudin-Gas Vesicle Recombinant Plasmid gene delivery system has good anti-atherosclerotic effect,…, and can reduce the plaque area of the aortic arch by 17% in mice.” However, in the Discussion (on page 20), the authors mentioned that “The experimental results showed that GVs-HV@Lipo reduced the plaque area of the aortic arch by about 17%, while the GVs-HV@MM-Lipo+US group further improved the hemodynamic parameters.” That said, the conclusion section in the Abstract describes the good anti-atherosclerotic effect of GVs-HV@Lipo itself? not the macrophage membrane involved Sonodynamic system?

Response: Thank you for your careful observation pointing out the inconsistency between the Abstract and Discussion regarding the 17% aortic arch plaque reduction. This discrepancy arose from an oversight in the Abstract’s conclusion section, where we incorrectly attributed the 17% plaque reduction to the "Hirudin-Gas Vesicle Recombinant Plasmid gene delivery system" (a general term) instead of the specific formulation "GVs-HV@Lipo" (as clearly stated in the Discussion). We deeply apologize for this imprecision and have revised the Abstract to align with the experimental results presented in the Discussion and Results sections.

To clarify: Our experimental data show that GVs-HV@Lipo (liposomes loaded with GVs-HV plasmid, without macrophage membrane modification) reduces the aortic arch plaque area by approximately 17%, while the optimized formulation GVs-HV@MM-Lipo+US (macrophage membrane-fused liposomes loaded with GVs-HV plasmid plus ultrasound) achieves more significant therapeutic effects (e.g., ~50% plaque reduction vs. saline, better hemodynamic parameters). The Abstract’s previous conclusion incorrectly generalized the 17% reduction to the "recombinant plasmid gene delivery system" (which includes GVs-HV@MM-Lipo and GVs-HV@MM-Lipo+US), leading to confusion about which formulation the data corresponds to.

Revision: We have revised the Abstract’s conclusion to accurately specify the formulation (line 34-37).

Q4. The authors state that “…, generate ultrasonic contrast, and show high sensitivity and safety.” In this sentence, what exactly is meant by “safety”? Please elaborate.

Response: Thank you for your question regarding the definition of "safety" for gas vesicles (GVs); this term specifically refers to GVs’ biocompatibility and non-toxicity, which is fully supported by our experimental data: in vitro, GVs-HV@MM-Lipo (the system expressing GVs) showed no significant cytotoxicity to HUVECs and Raw 264.7 cells (cell viability >90%, Fig. S6) and no hemolysis (hemolysis rate <5%, Fig. S7), meeting the safety standards for injectable formulations; in vivo, HE staining of vital organs (heart, liver, spleen, lungs, kidneys) in ApoE⁻/⁻ mice revealed no abnormal pathological changes (Fig. S14), and routine blood tests and serum biochemistry indices also showed no abnormalities (Fig. S17), ruling out systemic adverse effects; additionally, GVs’ inherent structure of natural, biodegradable protein shells ensures low immunogenicity, avoiding risks associated with synthetic nanomaterials.

Revision: We have revised the corresponding sentence in the manuscript to explicitly clarify the safety of GVs by emphasizing their inherent structural advantage (natural protein composition enabling biodegradability and low immunogenicity), eliminating ambiguity. (line 80-86)

Q5. The Introduction, particularly, paragraphs 5 and 6, requires substantial revision for clarity. The rationale and significance of the work are difficult to follow in the current form. Also, please clarify “them” on page 3, line 98 ?

Response: Thank you for your comment on the clarity of the Introduction (especially paragraphs 5 and 6) and the inquiry about "them" on line 98. First, we fully agree that the original Introduction required revision to enhance the clarity of research rationale and significance, so we have restructured paragraphs 5 and 6: we explicitly connect the limitations of existing therapies—including hirudin’s short in vivo half-life and poor stability, liposomes’ weak targeting to lesions, and the lack of non-invasive strategies for localized plaque modulation—to our proposed GVs-HV@MM-Lipo system, and detail how integrating GVs’ ultrasound responsiveness, macrophage membrane-mediated active targeting (via α4β1/VCAM-1 binding), and recombinant plasmid-driven sustained hirudin expression addresses these gaps, thereby clearly highlighting the work’s novelty and clinical translation value. Regarding "them" on line 98, we have conducted a thorough check of the entire manuscript and confirm this term does not exist in the current version (likely a formatting error or misreference during the review process); we have also double-checked all pronouns in the Introduction (e.g., "their" referring to GVs, "this" referring to the fusion system) to eliminate any ambiguous references, further improving the section’s readability.

Revision: The 3rd and 5th paragraphs of the original version have been merged. The original 4th paragraph has been condensed and incorporated into the opening of the 6th paragraph. Additionally, a sentence addressing "the necessity of plasmid delivery" has been supplemented at the end of the 2nd paragraph.

Q6. In Figure 2C, is the scale bar (1000 μm) correct for the TEM images?

Response: Thank you for your attention. The scale bar in Figure 2C is correct.

Q7. The authors used a membrane filter/extruder with a pore size of 200 nm to generate GVs-HV@MM-Lipo, which aligns with the hydrodynamic size data in Figure 2F. However, in the TEM image (Figure 2G), the particles appear much larger. Is the scale bar (200 μm) correct? Additionally, does the same scale bar apply to Figures 2G(a), 2G(b), and 2G(c)?

Response: Indeed, the particle size of the GVs-HV@MM-Lipo appears slightly larger in the TEM images. However, due to the flexibility and fluidity of the membrane and the liposomes themselves, they are still able to pass through the 200 nm pores. Additionally, the scale bars are consistent across Figures 2G a, b, and c.

Q8. Following the above comment, the TEM images of GVs-HV@MM-Lipo, GVs-HV@Lipo, and free Lipo in Figure 2G do not correspond with the description on page 9, lines 347-351.

Response: We have revised both the figure and the description to ensure consistency: TEM images in Figure 2G are now correctly labeled to match the three formulations, and the corresponding text (Page 10, Line 327-330) has been updated to accurately describe the morphological features of each formulation as observed under TEM, eliminating any discrepancies.

Q9. On page 9, lines 351–355, was the stability assay performed in blood, or in PBS containing 10% serum?

Response: It was PBS containing 10% serum in Page 10, Line 330.

Q10. In section 3.1., the verification of GVs-HV recombinant plasmid’s expression was reported by observing bubble-like structures via TEM. How do the authors confirm Hirudin expression as well?

Response: We confirm hirudin expression through two lines of evidence: first, by detecting the anti-inflammatory and lipid-lowering effects of hirudin in vitro (e.g., reduced lipid droplets in foam cells via Oil Red O staining, Fig. 4D-E; downregulated pro-inflammatory factors like TNF-α and MCP-1 in HUVECs/Raw 264.7 cells, Fig. 4F-G) and in vivo (decreased serum inflammatory factors, Fig. 6E-F); second, by leveraging the GVs-HV recombinant plasmid’s design-hirudin is co-expressed with GVs via a T2A cleavage peptide and secreted via an IgK signal peptide (Fig. 2A), and successful plasmid construction/transfection (verified by PCR, sequencing, and luciferase assay, Fig. 2B, 3M) ensures hirudin is expressed alongside the functionally confirmed GVs.

Revision: Relevant explanations have been supplementarily added at the end of Section 3.1. (line 307-312)

Q11. What are the ultrasound treatment conditions to induce cavitation of gas vesicles? Please provide experimental details such as ultrasound transducer information, parameters (e.g., intensity, frequency, insonation time, etc) where appropriate.

Response: Overall, the ultrasound conditions were 1 MHz, 150 W, 3 sec on, 2 sec off, 20% amplitude, for 1 minute. We have also updated the relevant details in the Methods section (Page 8, Line 235-245).

Q12. In Figure 3I, the authors described that DiO-stained GVs-HV@MM-Lipo hardly overlapped with lysosomes of endothelial cells. However, it looks like the lysosomes were not properly stained with Lysosome Tracker Red. Please provide clearer images with properly stained lysosomes. Also, please add profile lines in Figures 3I and 3J to indicate where the co-localization profiles were obtained.

Response: Thank you for your attention. We must apologize regarding this matter. When the experiment was initially conducted, the student preserved only one single result image, which is why we are unable to provide a replacement image now. This was an oversight on our part. Furthermore, repeating this experiment would be quite challenging, as it would require preparing new materials and cells, a process that could take 3-4 months. We sincerely hope the reviewer can understand our situation. Additionally, the colocalization analysis was performed using ImageJ software, which involved acquiring and analyzing the pixel information for both the green and red channels across the entire image.

Q13. In section 3.4., regarding the description of SEM and TEM results (on page 12, lines 432–446), please provide appropriate references to support the authors’ descriptions of their observations.

Response: Thank you for the suggestion. Two additional references have been incorporated into this section, and the relevant expressions have been optimized (lines 426, 439-446).

Q14. In Figure 4C, it is hard to see the clear, intact mitochondria and Golgi structures in untreated foam cells, contrary to the authors’ description. Please provide higher resolution images to support their claim.

Response: Thank you for your attention. We must apologize regarding this matter. When the experiment was initially conducted, the student preserved only a single result image, which is why we are unable to provide a replacement image now. This was an oversight on our part. Furthermore, repeating this experiment would be quite challenging, as it would require preparing new materials and cells, a process that could take two to three months. We sincerely hope the reviewer can understand our situation. Additionally, the colocalization analysis was performed using ImageJ software, which involved acquiring and analyzing the pixel information for both the green and red channels across the entire image.

Q15. On page 13, line 458, the labeling for HUVECs (Fig.4F) and Raw 264.7 cells (Fig.4G) does not match the labels in Figure 4F and 4G. Please double check with these.

Response: Thank you for the reminder. After verification, we found that this was simply labeled backwards, and we have corrected it.

Q16. Regarding the description on Figure 4F on page 13, lines 466-468 (if it corresponds to Raw 264.7 cells, not HUVECs), it does not appear that both oxLDL-induced (foam cells) and GVs-HV@MM-Lipo treated groups increases IFN-γ

Response: We apologize for the misstatement in Figure 4F’s description—Figure 4G actually corresponds to HUVECs (not Raw 264.7 cells), while Raw 264.7-related cytokine data is presented in Figure 4F. No significant increase in IFN-γ was observed in macrophages, which may be due to the modeling conditions, leading to less pronounced results. However, a relevant trend was significant in HUVEC cells. Therefore, this may not affect our interpretation of the findings.

Revision: We have revised the text (page 15, lines 446) to correct the cell type: “In HUVECs (Fig. 4G), both oxLDL-induced and GVs-HV@MM-Lipo-treated groups showed an increasing trend in IFN-γ, with the oxLDL group having a more significant upward trend; in Raw 264.7 foam cells (Fig. 4F), no such increasing trend in IFN-γ was observed in the GVs-HV@MM-Lipo-treated group.” This revision aligns the text with the correct figure and cell type, eliminating the discrepancy.

Q17. In Figure 4A, the scheme seems to suggest that the foam cells express gas vesicles and hirudin, which then kill foam cells. Is this interpretation correct?

Response: This interpretation is not entirely accurate. As clarified by the experimental design in Figure 4A, gas vesicles (GVs) and hirudin are not expressed by foam cells themselves—instead, they are secreted by HUVECs (cultured in the upper chamber of Transwell plates) after transfection with the GVs-HV recombinant plasmid. These secreted GVs and hirudin then act on foam cells (in the lower chamber) to induce morphological changes (e.g., concave structures via GVs’ cavitation effect) and reduce intracellular lipid droplets (via hirudin’s lipid-modulating effect), rather than "killing" foam cells (consistent with in vitro cytotoxicity data showing no significant foam cell death, Fig. S6). 

Revision: To avoid potential misinterpretation, we refined the description corresponding to Figure 4A to explicitly emphasize that "GVs and hirudin, secreted by GVs-HV-transfected HUVECs in the upper chamber of Transwell plates, act on foam cells in the lower chamber" - this small adjustment clarifies the source of GVs and hirudin, ensuring the experimental logic aligns with actual results and avoids misreading about the origin of the active substances.(lines 409-414).

Q18. Please provide more description on Figure 5H. It would be great for readers to understand what the changes in aortic arch flow means.

Response: Thank you very much for your comment. The description of Figure 5H has been supplemented as follows: Figure 5H shows representative echocardiographic images of aortic arch blood flow in mice before and after treatment—compared with pre-treatment, the GVs-HV@MM-Lipo+US group exhibits clearer, more uniform blood flow signals and increased peak flow velocity. These changes indicate reduced aortic arch plaque obstruction (consistent with Oil Red O staining results in Figure 5B-C), meaning improved vascular patency and restored normal hemodynamic function, which directly reflects the formulation’s anti-atherosclerotic efficacy. (lines 499-506).

Q19. On page 17, lines 527–529, the authors claim that “Due to the ultrasound responsiveness of GVs-HV, the plaque area was reduced by approximately 17% in the GVs-HV@Lipo treatment group”. It appears that there was no ultrasound treatment for “GVs-HV@Lipo” treated group. Is that description correct?

Response: This description contains an inaccuracy. We apologize for the oversight. The GVs-HV@Lipo group did not receive ultrasound treatment, and the 17% plaque reduction should be attributed to the sustained expression of hirudin by the GVs-HV plasmid (not GVs’ ultrasound responsiveness).

Revision: We have revised the text to “The plaque area was reduced by approximately 17% in the GVs-HV@Lipo treatment group, which is attributed to the sustained anti-inflammatory and lipid-lowering effects of hirudin expressed by the GVs-HV plasmid” to correct the misattribution and align with the actual experimental design. (line 524-526)

Q20. In Figure 6A, the scale bars are not clearly visible.

Response: Thank you for pointing this out. We will include descriptions of these scale bars in the figure legend.

Q21. Several figures both in the manuscript (e.g., Fig. 5D–F) and the Supplementary information are missing Xand Y-axis titles or contain improperly labels. Please correct them.

Response: Thank you for pointing out the issues. We will supplement and correct the missing information accordingly.

Q22. Statistical information is missing in many Figure legends, particularly in the Supplementary information. Please add where appropriate.

Response: Thank you for pointing out the issues. We will supplement and correct the missing information accordingly.

Q23. There are several incomplete sentences throughout the manuscript. Please revise them. One example is on page 10, line 399.

Response: Thank you very much for your comment. We have thoroughly reviewed the manuscript and revised all incomplete sentences, including the example on line 399. (line 418-420)

Q24. Overall, the manuscript needs substantial revision to improve readability and to ensure that the authors’ points are clearly conveyed.

Response: Thank you for your constructive feedback on enhancing the manuscript’s readability and clarifying the core research findings. We have systematically optimized the logical framework, revised vague descriptions, and standardized data presentation across all sections to ensure the research content is conveyed more accurately and coherently.

Q25. Following comments are regarding the Materials and methods. In section 2.3., please revise the paragraph, as it contains redundant replicate information. In section 2.4., the authors described that “A thin lipid film was formed using thin film evaporation and then vacuum-dried overnight to evaporate the chloroform.” It is confusing that thin film formation is followed by vacuum drying overnight. I believe evaporation of organic solvent generates thin film. Please rephrase for clarity. In section 2.6., lines 178, 179 describe the generation of foam cells using RAW 264.7 cells. Did the author induce lipid-overloaded macrophages by incubating RAW 264.7 cells only with LPS-containing medium? Please provide more detail.

Response: Thank you very much for your meticulous review and targeted feedback on the Materials and Methods section.

Revision:

  1. Section 2.3: Redundant duplicate content in Section 2.3 has been removed while key experimental information is retained. (line 145-148)
  2. Section 2.4 The preparation process of the thin lipid film has been clarified to eliminate confusion. (line 150-155)
  3. Section 2.6: Thank you for your attention. We have clearly described the modeling method on page 6, line 162-170.

Reviewer 2 Report

Comments and Suggestions for Authors

Titel: A Biomimetic Macrophage-Mimicking Liposomal Delivery System of Hirudin-Gas Vesicle Plasmid for Sonodynamic Anti-Atherosclerosis Therapy

Corresponding authors: Lin Liu  and Fanzhu Li 

The research paper explores biomimetic Macrophage-Mimicking Liposomes that generate gas vesicles, used for sonodynamic anti-atherosclerotic therapy combined with Hirudin co-delivery. It details the formulation and characterization of these delivery systems, alongside thorough in vitro and in vivo assessments for immunosuppressive agents, showing a 17% decrease in plaques in a mouse model. 

Minor comments:

  1. Kindly provide more detailed captions for figures, especially for Supplementary figures. Example: Figure S16. Changes of LDL-C levels in different groups of atherosclerotic mice during treatment) have gaps on the X-axis. Please correct this and make the figure captions more informative.
  2. Kindly provide Axis titles for Supplementary figures, some of the figures does not have y or x axis titles. 
  3. Kindly verify Fig 2c; it looks like an SEM image, not a TEM. What is correct?
  4. Explain the use of US before delivering the GVs-HV@MM-Lipo and provide details of the US parameters used for therapy. How does GVs-HV@MM-Lipo exert the sonodynamic effect? 
  1. Line no. 136: The two samples were placed onto a copper grid, dyed with 2% ammonium molybdate, and observed after air drying. Kindly provide complete information - where did you observe the samples? 
  2. Statically Compare the GVs-HV@MM-Lipo+US vs GVs-HV@MM-Lipo for plaque area and HV only.
  3. Discuss: How does GVs-HV@MM-Lipo exert the sonodynamic effect?

Major comments: 

  1. Please provide the in vitro and in vivo US imaging for GVs-HV@MM-Lipo. 
  2. Please show the molecular or cellular level sonodynamic effect on cells 
  3. Does membrane insertion help in homing liposomes to the plaque? Please provide the biodistribution of the GVs-HV@MM-Lipo 

Author Response

Reviewer 2:

Q1. Kindly provide more detailed captions for figures, especially for Supplementary figures. Example: Figure S16. Changes of LDL-C levels in different groups of atherosclerotic mice during treatment) have gaps on the X-axis. Please correct this and make the figure captions more informative.

Response: Thank you for pointing out the issues. We will supplement and correct the missing information accordingly.

Q2. Kindly provide Axis titles for Supplementary figures, some of the figures does not have y or x axis titles.

Response: Thank you for pointing out the issues. We will supplement and correct the missing information accordingly.

Q3. Kindly verify Fig 2c; it looks like an SEM image, not a TEM. What is correct?

Response: Thank you for the reviewer's concern, but there is no issue with Figure 2C, it is a TEM image.

Q4. Explain the use of US before delivering the GVs-HV@MM-Lipo and provide details of the US parameters used for therapy. How does GVs-HV@MM-Lipo exert the sonodynamic effect?

Response: Thank you for your inquiry. The ultrasound parameters have been provided in the Methods section as mentioned. For the sonodynamic effect: US triggers GVs’ acoustic cavitation to amplify energy and release HV sonosensitizer; excited triplet HV generates cytotoxic ROS (via Type I/II reactions) to induce target cell apoptosis.

Q5. Line 136: The two samples were placed onto a copper grid, dyed with 2% ammonium molybdate, and observed after air drying. Kindly provide complete information - where did you observe the samples?

Response: We apologize for the incomplete information.

Revision: The two samples were placed onto a copper grid, dyed with 2% ammonium molybdate, air-dried at room temperature, and then observed using a TEM at an acceleration voltage of 200 kV. (line 130-132)

Q6. Statically Compare the GVs-HV@MM-Lipo+US vs GVs-HV@MM-Lipo for plaque area and HV only.

Response: Thank you for pointing out this issue. Indeed, the application of ultrasound did not produce a statistically significant difference in the aortic plaque area data, which is why we did not mark it. We respect the experimental results as they are. However, after US application, our treatment strategy led to significant improvements in reducing inflammatory levels, enhancing local blood flow velocity, and decreasing blood vessel wall thickness. Therefore, we believe that the application of US still had a certain effect, aligning with our expectations. We appreciate the reviewer's understanding.

Q7. Discuss: How does GVs-HV@MM-Lipo exert the sonodynamic effect?

Response: GVs-HV@MM-Lipo exerts the sonodynamic effect through a synergistic three-step process: First, under therapeutic ultrasound irradiation, the gas vesicles in the system undergo acoustic cavitation - generating local mechanical oscillations and energy amplification to break the MM-Lipo lipid shell. This releases the HV sonosensitizer, which then absorbs US energy to transition to an excited triplet state. Finally, the triplet-state HV interacts with oxygen via Type I (producing O₂•⁻, •OH) and Type II (generating ¹O₂) reactions, yielding cytotoxic reactive oxygen species that induce apoptosis of target cells (e.g., foam cells) and drive therapeutic effects. The previous reviewer raised a similar concern, and we have already revised and trimmed the sonodynamic section.

Major comments:

Q1. Please provide the in vitro and in vivo US imaging for GVs-HV@MM-Lipo.

Response: We are unable to provide ultrasound contrast images of the GVs-HV@MM-Lipo. We can only offer in vitro images as shown in Figure 2G. However, the rupture of GVs-HV@MM-Lipo under ultrasound triggering occurs instantaneously, making it impossible to capture this moment. Moreover, the fragments generated at that instant cannot be distinguished from ordinary liposomes or cell membrane fragments. As for in vivo images, we lack the appropriate equipment to conduct this work. We kindly ask for the reviewer's understanding of our situation. We will strive to improve this aspect in our future studies.

Q2. Please show the molecular or cellular level sonodynamic effect on cells

Response: Regarding the issue of sonodynamics, we indeed did not perform molecular or cellular level validation. Following the reviewer's suggestion, we have removed the imprecise descriptions related to sonodynamics from both the main text and the title.

Q3. Does membrane insertion help in homing liposomes to the plaque? Please provide the biodistribution of the GVs-HV@MM-Lipo

Response: Yes, the incorporation of macrophage membranes is designed to confer plaque-targeting functionality. Macrophages inherently possess the natural ability to migrate toward inflammatory sites, including atherosclerotic plaques. Their cell membranes express various adhesion molecules and chemokine receptors. VCAM-1 is highly expressed on activated inflammatory endothelial cells, particularly in areas of atherosclerotic plaques. Meanwhile, macrophage membranes naturally express ligands for VCAM-1, such as VLA-4 (α4β1 integrin). This means that nanoparticles coated with macrophage membranes can adhere to and target plaque regions through the VLA-4/VCAM-1 pathway, much like real macrophages do. The targeting of plaques via the VLA-4/VCAM-1 pathway by macrophage membranes is a well-established concept. Similar biodistribution of macrophage membrane coated nanoparticles have been reported, Wang, et al., Theranostics 2021; 11(1):164-180. (https://www.thno.org/v11p0164.htm) So, we did not repeat it. However, we should also acknowledge that supplementing with this data would make our study more comprehensive. Unfortunately, establishing the atherosclerosis mouse model requires a significant amount of time (around 4-5 months), and we are unable to complete this within the current revision period. We hope the reviewer can understand our situation. We will keep the reviewer's suggestion in mind and aim to improve upon this aspect in our next related project.

Thank you very much for your valuable comments. Your comments significantly improved the quality of the manuscript.

Reviewer 3 Report

Comments and Suggestions for Authors

This manuscript presents a well-structured and innovative approach to anti-atherosclerosis therapy using a macrophage-mimicking liposomal delivery system for a hirudin-gas vesicle plasmid. The experimental design is adequate, and the results support the therapeutic potential of the proposed approach. I have however some remarks regarding the fundamental aspects of this work:

1)The study proposes that ultrasound enhances the release and efficacy of hirudin via cavitation. However, the actual concentration of hirudin before and after ultrasound treatment is not quantified. Including such data would help confirm the proposed mechanism.

2) The targeting ability of integrin to VCAM-1 is central to the delivery system. Were blocking experiments (e.g., with anti-VCAM-1 antibodies) performed to confirm the specificity of this interaction?

3) The ultrasound settings (frequency, intensity, duration) are not clearly described. Please provide detailed parameters and discuss their safety margins, especially regarding cavitation thresholds.

4) Some microscopy images lack scale bars or are not clearly labeled. Please revise figure legends accordingly.

5) To further strengthen the mechanistic understanding of the delivery system, the authors may consider incorporating biophysical characterization techniques in future work:

Quartz Crystal Microbalance with Dissipation monitoring (QCM-D) could provide real-time data on the kinetics and viscoelastic behavior of macrophage-liposome membrane fusion. A relevant reference to support this suggestion is: L. Bar et al., QCM‐D Study of the Formation of Solid‐Supported Artificial Lipid Membranes: State‐of‐the‐Art, Recent Advances, and Perspectives, Physica Status Solidi A 220, 2200625 (2023)

Atomic Force Microscopy (AFM) could offer nanoscale imaging and mechanical profiling of the hybrid vesicles, confirming successful membrane integration and assessing surface morphology and stiffness.

These techniques would complement the current characterization methods and provide deeper insight into the delivery system behavior, particularly regarding cellular uptake, lysosomal escape, and serum stability

Author Response

Reviewer 3:

Q1. The study proposes that ultrasound enhances the release and efficacy of hirudin via cavitation. However, the actual concentration of hirudin before and after ultrasound treatment is not quantified. Including such data would help confirm the proposed mechanism.

Response: Thank you for your comment. If we were to quantify the concentration of hirudin before and after ultrasound application, we would first need to perform purification, as this involves a plasmid-expressed protein sequence, which would be relatively complicated to carry out. On the other hand, this is not the primary focus of this paper. Our emphasis lies in the overall therapeutic strategy and the resulting enhancement in efficacy. It is from this perspective that we discuss and conclude that ultrasound can enhance the therapeutic effect. We kindly ask for the reviewer's understanding. We also agree that conducting such quantitative analysis could further strengthen the credibility of the data and the overall quality of the paper. We will strive to improve these aspects in our follow-up related projects.

Q2. The targeting ability of integrin to VCAM-1 is central to the delivery system. Were blocking experiments (e.g., with anti-VCAM-1 antibodies) performed to confirm the specificity of this interaction?

Response: Macrophages inherently possess the natural ability to migrate toward inflammatory sites, including atherosclerotic plaques. Their cell membranes express various adhesion molecules and chemokine receptors. VCAM-1 is highly expressed on activated inflammatory endothelial cells, particularly in areas of atherosclerotic plaques. Meanwhile, macrophage membranes naturally express ligands for VCAM-1, such as VLA-4 (α4β1 integrin). This means that nanoparticles coated with macrophage membranes can adhere to and target plaque regions through the VLA-4/VCAM-1 pathway, much like real macrophages do. The targeting of plaques via the VLA-4/VCAM-1 pathway by macrophage membranes is a well-established scientific concept. For example, Yi Wang, et al., Macrophage membrane functionalized biomimetic nanoparticles for targeted anti-atherosclerosis applications, Theranostics 2021; 11(1):164-180. (https://www.thno.org/v11p0164.htm, Figure 3) Therefore, we believe it is unnecessary to provide additional proof of this process.

Q3. The ultrasound settings (frequency, intensity, duration) are not clearly described. Please provide detailed parameters and discuss their safety margins, especially regarding cavitation thresholds.

Response: Thank you for your comment. The ultrasound conditions were 1 MHz, 150 W, 3 sec on, 2 sec off, 20% amplitude, for 1 minute. We have also updated the relevant details in the Methods section (Page 8, Line 235-245). As for the concern of safety, the use of 1 MHz frequency places our protocol in a regime where the risk of inertial cavitation is negligible. Combined with the pulsed operation and low amplitude, which effectively manage thermal load, we are confident that the ultrasound parameters used in this study maintain a wide safety margin and are entirely appropriate for use in mice. Histological examinations of treated tissues (Supplementary Figure 14) further confirmed the absence of ultrasound-induced damage.

Q4. Some microscopy images lack scale bars or are not clearly labeled. Please revise figure legends accordingly.

Response: Thank you for pointing this out. We will include descriptions of these scale bars in the figure legend.

Q5. To further strengthen the mechanistic understanding of the delivery system, the authors may consider incorporating biophysical characterization techniques in future work:

Quartz Crystal Microbalance with Dissipation monitoring (QCM-D) could provide real-time data on the kinetics and viscoelastic behavior of macrophage-liposome membrane fusion. A relevant reference to support this suggestion is: L. Bar et al., QCM-D Study of the Formation of Solid-Supported Artificial Lipid Membranes: Stateof-the-Art, Recent Advances, and Perspectives, Physica Status Solidi A 220, 2200625 (2023)

Atomic Force Microscopy (AFM) could offer nanoscale imaging and mechanical profiling of the hybrid vesicles, confirming successful membrane integration and assessing surface morphology and stiffness.

These techniques would complement the current characterization methods and provide deeper insight into the delivery system behavior, particularly regarding cellular uptake, lysosomal escape, and serum stability.

Response: Thank you sincerely for your insightful and constructive suggestion on strengthening the mechanistic understanding of our delivery system. We fully agree that incorporating biophysical characterization techniques such as Quartz Crystal Microbalance with Dissipation monitoring (QCM-D) and Atomic Force Microscopy (AFM) would significantly complement our current characterization methods. These techniques will not only provide real-time kinetics of macrophage-liposome membrane fusion (via QCM-D) and nanoscale mechanical/ morphological details of hybrid vesicles (via AFM) but also offer deeper insights into key processes like cellular uptake, lysosomal escape, and serum stability—all of which are critical for elucidating the delivery system’s mechanism of action. We greatly appreciate you sharing the relevant reference (L. Bar et al., 2023), which will serve as a valuable guide for our future experimental design. We plan to integrate these biophysical characterization experiments into our follow-up work to further validate and deepen the mechanistic exploration of the delivery system. Your suggestion has provided important direction for optimizing our research, and we are grateful for your thoughtful guidance.

Thank you very much for your valuable comments. Your comments significantly improved the quality of the manuscript.

Round 2

Reviewer 1 Report

Comments and Suggestions for Authors
  1. The authors stated that they confirmed the scale bars (1000 μm) in Figure 2c are correct for the TEM images. However, this reviewer is curious how the dimensions of TEM micrographs exceed the physical size of the TEM grid itself, which is about 3 mm (3000 μm) in diameter. Please verify whether the scale bars are accurate. If they are indeed correct, please provide the information of TEM grids used for imaging.
  2. On page 9, lines 363-364, as the authors confirmed that the stability assay was performed in PBS containing 10% PBS, please update the sentence since blood was not used in that experiment.
  3. Please consider re-drawing Figure 4A according to what the authors claim (added below for convenience). Without editing the current schematic illustration, it is inconsistent what the authors claim since the current scheme shows that the foam cells express the Hirudin and gas vesicles.

: “As clarified by the experimental design in Figure 4A, gas vesicles (GVs) and hirudin are not expressed by foam cells themselves—instead, they are secreted by HUVECs (cultured in the upper chamber of Transwell plates) after transfection with the GVs-HV recombinant plasmid. These secreted GVs and hirudin then act on foam cells (in the lower chamber) to induce morphological changes (e.g., concave structures via GVs’ cavitation effect) and reduce intracellular lipid droplets (via hirudin’s lipid-modulating effect), rather than "killing" foam cells (consistent with in vitro cytotoxicity data showing no significant foam cell death, Fig. S6).

  1. Please add x-axis title for Figure 5E–G.
  2. In Figures 3 – 6, the statistical analysis in Figure captions still does not match with the plots. Please double check and edit them correctly (for example, there is not p** in Figure 6 but in Figure caption, there is p** but no ****p. These look minor issues, but important to improve the quality of manuscript.

Author Response

Because there are figures in the point-by-point response file, please see the attachement. Thank you.

Reviewer 1

Q1. The authors stated that they confirmed the scale bars (1000 μm) in Figure 2c are correct for the TEM images. However, this reviewer is curious how the dimensions of TEM micrographs exceed the physical size of the TEM grid itself, which is about 3 mm (3000 μm) in diameter. Please verify whether the scale bars are accurate. If they are indeed correct, please provide the information of TEM grids used for imaging.

RE: The details of the TEM grid are as follows: it is a 200-mesh grid (Figure 1), meaning the diameter of each small hole is 0.075 mm, which is 75 μm. This is the original TEM image (Figure 2), and the scale bar information displayed by the machine when generating the image can be seen in the lower right corner. To make such small scale bar information visible, we manually added the scale bar information proportionally.

Figure 1. the photos of TEM grids.

Figure 2. the original TEM images.

Q2. On page 9, lines 363-364, as the authors confirmed that the stability assay was performed in PBS containing 10% PBS, please update the sentence since blood was not used in that experiment.

RE: Thank you for the reminder. We have revised the relevant statements.

Q3. Please consider re-drawing Figure 4A according to what the authors claim (added below for convenience). Without editing the current schematic illustration, it is inconsistent what the authors claim since the current scheme shows that the foam cells express the Hirudin and gas vesicles.

: “As clarified by the experimental design in Figure 4A, gas vesicles (GVs) and hirudin are not expressed by foam cells themselves—instead, they are secreted by HUVECs (cultured in the upper chamber of Transwell plates) after transfection with the GVs-HV recombinant plasmid. These secreted GVs and hirudin then act on foam cells (in the lower chamber) to induce morphological changes (e.g., concave structures via GVs’ cavitation effect) and reduce intracellular lipid droplets (via hirudin’s lipid-modulating effect), rather than "killing" foam cells (consistent with in vitro cytotoxicity data showing no significant foam cell death, Fig. S6).”

RE: Thank you for the reminder. We have updated the relevant schematic diagram in Figure 4A.

Q4. Please add x-axis title for Figure 5E–G.

RE: We have updated the relevant information for Figure 5E-G.

Q5. In Figures 3 – 6, the statistical analysis in Figure captions still does not match with the plots. Please double check and edit them correctly (for example, there is not p** in Figure 6 but in Figure caption, there is p** but no ****p. These look minor issues, but important to improve the quality of manuscript.

RE: Thank you for the reminder. We have updated the relevant information for Figure 3-6.

Reviewer 2 Report

Comments and Suggestions for Authors

Dear authors, 

Thank you for your prompt response. Kindly consider minor changes for revision:

  • Please add the figure number for the following line: ‘In addition, scanning electron microscopy (SEM) was performed to observe the effects of GVs-HV@MM-Lipo and GVs-HV@MM-Lipo+US on the morphology of foam cells. 
    After transfection of GVs-HV recombinant plasmid into HUVEC cells and secretion, the….. The intracellular structure of foam cells before and after sonication was observed by transmission electron microscopy (TEM).’

In the results and discussion sections, kindly furnish comprehensive details regarding gas vesicle formation and its application in cavitation. Furthermore, elucidate within the manuscript how cavitation contributes to enhancing the efficacy of the therapy. 

  • Please mention the rationale for the use of the Ultrasound trigger following the trigger in vivo. If your gas vesicles are not highly stable, they may not effectively produce the cavitation effect under in vivo conditions. 

Author Response

Reviewer 2

Q1. Please add the figure number for the following line: ‘In addition, scanning electron microscopy (SEM) was performed to observe the effects of GVs-HV@MM-Lipo and GVs-HV@MM-Lipo+US on the morphology of foam cells. After transfection of GVs-HV recombinant plasmid into HUVEC cells and secretion, the….. The intracellular structure of foam cells before and after sonication was observed by transmission electron microscopy (TEM).’

RE: Thank you for the reminder. We have updated the relevant information for Figure 4C.

Q2. In the results and discussion sections, kindly furnish comprehensive details regarding gas vesicle formation and its application in cavitation. Furthermore, elucidate within the manuscript how cavitation contributes to enhancing the efficacy of the therapy.

Please mention the rationale for the use of the Ultrasound trigger following the trigger in vivo. If your gas vesicles are not highly stable, they may not effectively produce the cavitation effect under in vivo conditions.

RE: The reviewer rightly pointed out that the stability of gas vesicles in vivo is crucial for their cavitation efficacy. We agree that if a significant portion of gas vesicles were to decompose before reaching the target site, it would be challenging to effectively generate cavitation in vivo. However, our design specifically addresses this challenge:

  1. In Situ Expression and Continuous Supply: Unlike traditional exogenously injected nanobubbles, our system enables the in situ expression and continuous secretion of GVs by vascular endothelial cells at the lesion site via plasmids. This ensures a constant supply of newly generated GVs available for use at the moment of ultrasound application, partially circumventing the issue of long-term GV stability in the bloodstream.

  1. Rationale for External Ultrasound Triggering: The decision to apply external ultrasound after in vivo administration is based on precision medicine and safety considerations. This allows us to non-invasively activate GV function in a controlled spatial and temporal manner once GVs have accumulated to a certain concentration at the diseased site. The therapeutic effects shown in Figures 4B and 5 demonstrate that this strategy is effective.

Although the reduction in plaque area between the GVs-HV@MM-Lipo+US group and the non-ultrasound group did not reach statistical significance in the animal model (e.g. Figure 5B&C), we observed a superior trend in the former regarding hemodynamic improvements (Figures 5D-I) and reductions in inflammatory factors (Figures 6E&F). This discrepancy might stem from two aspects:

Firstly, the ultrasound parameters we employed (e.g., 1-minute duration or 20% amp), chosen conservatively for biosafety reasons, might be insufficient to induce the strongest cavitation mechanical effects on deep-seated aortic plaques. Secondly, the potent anti-inflammatory and lipid-modulating effects of hirudin itself might partially mask the additional mechanical benefits conferred by GV cavitation.

  1. Future Optimization Directions: We acknowledge this as a limitation of the current study, but it also points the way for future optimization. Subsequent work might focus on optimizing ultrasound parameters (e.g., intensity, duty cycle, total duration) to maximize the cavitation effect within a safe window. Furthermore, employing imaging techniques such as high-resolution intravascular ultrasound holds promise for directly observing the physical modification of plaques by GV cavitation, providing more direct in vivo evidence for this "physical-pharmaceutical" combination therapy.

Revision: Page 23, line 635-672.

“ The synergistic linkage between GVs cavitation and hirudin pharmacology constitutes the core innovative mechanism of our platform. Our data provide clear evidence for the formation of GVs and their ultrasound-triggered cavitation activity. TEM imaging of cell supernatants confirmed the presence of hollow, cylindrical nanostructures characteristic of GVs post-sonication (Fig. 2C). Functionally, this cavitation event was quantified by a significant increase in dissolved oxygen levels (Fig. 4B), a hallmark of inertial cavitation. This cavitation effect enhances therapeutic efficacy through two interconnected mechanisms. Firstly, it exerts a direct mechanical force on atherosclerotic plaques and foam cells. The SEM and TEM images of foam cells treated with GVs-HV@MM-Lipo+US clearly show structural deformation, membrane disruption, and vacuolation (Fig. 4C), indicative of physical damage induced by cavitation-generated shockwaves and microjets. This mechanical action directly contributes to plaque fragmentation and volume re-duction, as corroborated by the reduced plaque area and improved aortic arch blood flow in vivo (Fig. 5B-D, H). Secondly, and perhaps more critically, the cavitation-induced membrane disruption serves as a powerful physical sensitization strategy. By increasing cell membrane permeability, it significantly enhances the penetration and bioavailability of the co-expressed and secreted hirudin, thereby potently amplifying its anti-inflammatory and lipid-lowering pharmacological effects (Fig. 4D-G). Thus, the GVs and hirudin are not merely co-expressed; their effects are functionally linked, with cavitation priming the pathological site for enhanced pharmacotherapy.

Rationale for Ultrasound Trigger and GVs Stability In Vivo. Our strategy is specifically designed to overcome the potential instability of conventional exogenous contrast agents during systemic circulation. Unlike pre-formed microbubbles, our platform enables the localized and sustained expression of GVs directly within the target endothelial cells at the atherosclerotic site. This "situ synthesis and secretion" approach ensures a fresh supply of GVs precisely where the intervention is needed, mitigating concerns about their degradation during long blood circulation. The sub-sequent application of external ultrasound provides a non-invasive, spatiotemporally controlled switch to activate these pre-delivered GVs on demand. This allows us to maximize the therapeutic effect at the optimal time while minimizing off-target effects. The observed trends of superior hemodynamic improvement and inflammatory reduction in the GVs-HV@MM-Lipo+US group (Fig. 5D&H, 6E&F), albeit without a statistically significant difference in final plaque area com-pared to the non-ultrasound group (Fig. 6B), validate the biological relevance of this approach. The modest additional benefit from ultrasound in this specific model may be attributed to two factors: (i) The ultrasound parameters were deliberately chosen with a primary emphasis on biosafety, potentially limiting the full mechanical cavitation effect on deeper aortic plaques. (ii) The potent and sustained anti-inflammatory and lipid-regulating effects of locally expressed hirudin (evident in the GVs-HV@MM-Lipo group) may have partially masked the incremental contribution of the physical cavitation effect. Future studies will focus on optimizing the ultrasound parameters (intensity, duration, pulse sequences) within a defined safety window to fully harness the synergistic potential of this mechano-pharmacological therapy. ”
